# Target Speaker Extraction through Comparing Noisy Positive and Negative Audio Enrollments

**Shitong Xu**   **Yiyuan Yang**   **Niki Trigoni**   **Andrew Markham**
*Department of Computer Science, University of Oxford*
shitong.xu@cs.ox.ac.uk

## Abstract

Target speaker extraction focuses on isolating a specific speaker's voice from an audio mixture containing multiple speakers. To provide information about the target speaker's identity, prior works have utilized clean audio samples as conditioning inputs. However, such clean audio examples are not always readily available. For instance, obtaining a clean recording of a stranger's voice at a cocktail party without leaving the noisy environment is generally infeasible. Limited prior research has explored extracting the target speaker's characteristics from noisy enrollments, which may contain overlapping speech from interfering speakers. In this work, we explore a novel enrollment strategy that encodes target speaker information from the noisy enrollment by comparing segments where the target speaker is talking (Positive Enrollments) with segments where the target speaker is silent (Negative Enrollments). Experiments show the effectiveness of our model architecture, which achieves over 2.1 dB higher SI-SNRi compared to prior works in extracting the monaural speech from the mixture of two speakers. Additionally, the proposed two-stage training strategy accelerates convergence, reducing the number of optimization steps required to reach 3 dB SNR by 60%. Overall, our method achieves state-of-the-art performance in the monaural target speaker extraction conditioned on noisy enrollments. Our implementation is available at https://github.com/xu-shitong/TSE-through-Positive-Negative-Enroll .

## 1   Introduction

In the target speaker extraction task, the model is required to extract the target speaker's voice from a mixture of multiple speakers and ambient noise. To specify the characteristics of the target speaker, prior works have explored using conditional information of the target speaker in multiple modalities, including visual [1, 2], contextual [3, 4], or acoustic modality [5, 6, 7, 8]. Though prior works using conditional information in the acoustic modality have achieved significant good extraction performance, most of these works only considered using clean audio examples of the target speaker [9, 10, 11, 12]. This strong assumption prevents these models from performing well when only noisy audio examples are available. For example, consider a cocktail party, where the user meets a stranger who has not provided an audio sample before. To extract the target stranger speaker's voice from the noisy environment to assist clear conversation, the user will have to ask the speaker to step outside to record the clean audio enrollment. Enrolling target speakers in this way is often impractical in real-world applications.

Although prior works have attempted to perform target speaker extraction using noisy enrollments, they either still exploit the timesteps where *only* the target speaker is speaking in the noisy enrollment [13, 14], or assume that the user has participated in the audio mixture recording [15, 16], limiting extraction from arbitrary audio mixtures. In this work, we present a method for performing target speaker extraction conditioned on noisy audio enrollments where both the target speaker and the

39th Conference on Neural Information Processing Systems (NeurIPS 2025).

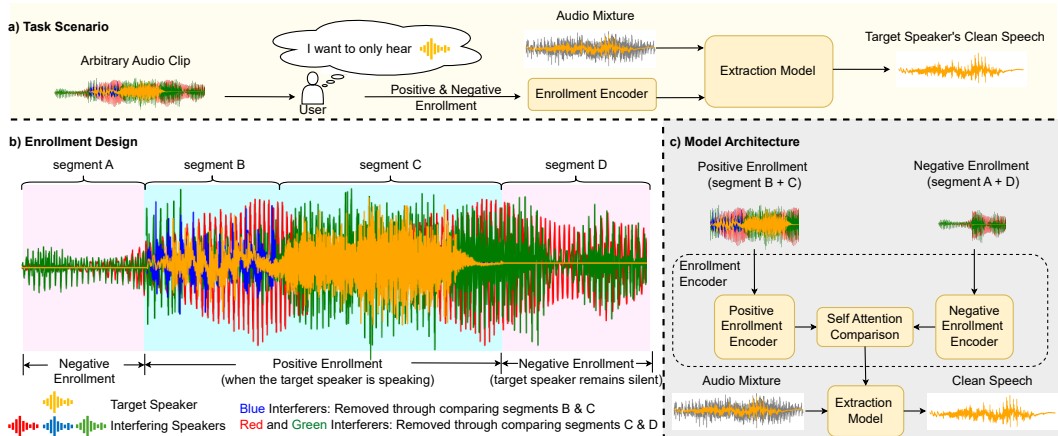

Figure 1: a) Task scenario explored in the work. Users identify a speaker of interest in an audio mixture, and labeling when the target speaker speaks (Positive Enrollment) or remains silent (Negative Enrollment) in the audio mixture. b) Decomposition of each speaker's voice in the audio mixture. Due to the stochasticity in human conversation, the interfering speaker will either remain silent in some of the segments in the Positive Enrollment or speak in the Negative Enrollment, leaving the target speaker the only speaker who talks throughout the Positive Enrollments but not in the Negative Enrollment. c) The model performs self-attention between the encoded Positive and Negative Enrollments to extract the target speaker's characteristic, which serves as the conditional information for the following extraction model. The model then extract the target speaker from the Audio Mixture.

interfering speakers overlap, as shown in Figure 1 (a). To resolve the ambiguity of which speaker in the noisy enrollment is the target speaker, we exploit the stochastic nature of different speakers' speaking patterns, which allows us to assume that speakers won't start and stop talking in perfect synchrony in a noisy conversational environment[1]. As shown in Figure 1 (b), based on this assumption, we structure the enrollment input as a combination of two segments: a Positive Enrollment (where the target speaker is speaking) and a Negative Enrollment (where the target speaker remains silent). During training, we encourage the model to encode the identity of the speaker who consistently speaks during Positive Enrollments and remains silent during Negative Enrollments. Interfering speakers, from the assumption above, will be inactive during parts of the Positive Enrollment (e.g., Blue speaker in segment C) and/or present during the Negative Enrollment (e.g., Red and Green speakers in segments A and D). By exploiting these temporal misalignments, the model can distinguish and encode the target speaker's characteristics, despite the presence of interfering speech and noise throughout both enrollments. We detail the problem formulation and the training pipeline to achieve this in Section 3.

By performing target speaker extraction in this manner, our model supports a broad range of real-world applications. Enrollment could easily be achieved in reality by pressing a button on an app or tapping an earbud to indicate a positive segment. Note that precise labeling is not required. Similarly, for offline speaker extraction (e.g., from an audio recording of a meeting), a few seconds of audio can be easily labeled as Positive or Negative samples.

In conclusion, our contributions include:

1. Exploring a novel enrollment strategy for target speaker extraction with noisy positive/negative enrollments and imprecise labels.

2. Design of a two-branch encoder and an associated two-stage training strategy for the task. Ablation experiments show that the proposed training method allows our model to achieve the same level of performance (3.0 dB SNR on the validation set) with 60% fewer optimization steps.

3. Demonstrate and discuss our model's performance across challenging and realistic application scenarios, including different numbers of interfering speakers, significant overlap between target and interfering speakers in enrollments, and inaccuracies in the positive and negative enrollment labeling.

---

[1]We quantitatively verify the feasibility of this assumption in Appendix A.

## 2 Related Work

### 2.1 Target Speaker Extraction in Challenging Scenarios

In target speaker extraction, prior works have explored using visual [1, 2], textural [3, 4], and audio examples [5, 6, 7, 8] of the target speaker as extraction conditions. Our work belongs to the third category, which learns the acoustic characteristics from given audio examples. Prior works in this category have attempted to improve extraction quality by modifying the fusion method [9, 17], leveraging multi-channel information in the audio mixture input [10, 18], and processing audio in the temporal domain [6, 7].

To improve the models' robustness and applicability in real-world applications, prior works have explored extracting multiple target speakers' speech simultaneously [19, 3, 20], transferring to extract speech for different languages [12], and disentangling irrelevant audio characteristics (e.g., reverberation effect) from the enrollment [21, 22, 23, 24, 25, 26]. In particular, Zhao et al. [11] addressed the problem of variable interfering speaker numbers in the audio mixture to be separated. In our experiment, we also show our model's generalizability to extract with different numbers of target and interfering speakers. Ranjan et al. [22] adopted curriculum learning to gradually increase the extraction difficulty. After a fixed number of epochs, samples with a lower signal-to-noise ratio (SNR) and higher similarity between target and interfering speakers are added to the training dataset. Our method adopts a similar training paradigm to pretraining the audio encoder. Both ours and the prior work from Ranjan et al. [22] show that the multi-stage training improves model convergence speed and leads to better final performance.

### 2.2 Target Audio Extraction Conditioned on Enrollments with Interfering Speakers

Though prior works have achieved significant progress in target audio extraction, most of these works assume the availability of clean audio enrollment. However, in real-world application scenarios, such enrollment examples are not necessarily available.

Among those prior works that attempted to extract target speech conditioned on noisy audio enrollments, TCE [15] explored the turn-taking dynamics in the conversation. TCE model accepts an audio encoding of the user's clean voice and performs extraction by considering the speakers who cross-talk with the user as the interfering speakers. However, TCE is limited to performing target speaker extraction when the user is participating in the conversation and does not allow specifying audio segments where the target speaker is present. A closely related work to ours is ADEnet [13], which also utilizes target speaker activity labeling to perform speech extraction. However, ADEnet only validated its performance on two-speaker mixtures with an average overlap ratio of 38.5%, where the enrollment contains mostly clean speech. As a result, it does not explore the challenges of unstable optimization and high overlap ratio between the target speaker and the interferers in the enrollment segment, which we explicitly address in our work. LookOnceToHear [16] leverages binaural audio to extract the target speaker's voice. During the enrollment stage, when the user is looking toward the target speaker, the model performs beamforming at a 90-degree azimuth to disambiguate the target speaker from the interferers talking simultaneously.

In comparison to the previous target speaker extraction methods, our model does not assume prior knowledge of any speaker in the mixed audio or the target speaker's spatial location in the enrollment stage. This allows our model to extract the target speaker from an arbitrary mono-channel sound mixture, and only use easily obtained binary labels of positive and negative audio segments in the temporal domain to extract the target speech. Besides, our model can flexibly perform extraction using either clean or noisy enrollments without retraining and get comparable performance with the previous method trained on clean enrollment, as shown in the Appendix J.

## 3 Method

### 3.1 Problem Formulation

In the target speaker extraction (TSE) task, the model first encodes the target speaker's characteristic from an enrollment input using an encoding branch. Conditioned on the encoding branch output, the model extracts the target speaker's voice from an Audio Mixture $a^M$. To allow the model to extract the target speaker conditioned on noisy enrollments, we formulate our enrollment as a pair of Positive Enrollment $a^P$ and Negative Enrollment $a^N$. After encoding the target speaker characteristic

$F(a^P, a^N)$ from the enrollment pair, the model extracts the target speaker's voice from the audio mixture $G(a^M, F(a^P, a^N))$.

To simulate the audio mixture in a noisy environment, we model the signal as the sum of an individual speaker's voice and the background noise. Thus, the noisy Positive Enrollment $a^P$, Negative Enrollment $a^N$, and the Audio Mixture $a^M$ are constructed as:

$$a^M = \sum_{i \in S^{IM} \cup \{t\}} a_i^M + n^M, \, a^P = \sum_{i \in S^{IE} \cup \{t\}} a_i^P + n^P, \, a^N = \sum_{i \in S^{IE}} a_i^N + n^N, \quad (1)$$

where $a_i^{\{P,N,M\}}$ represents the speech signal of speaker $i$ in the Positive, Negative, and Audio Mixture. $n^{\{P,N,M\}}$ represents the background noise in the three mixtures. $t$ represents the target speaker ID, and $S^{IE}, S^{IM}$ represent the interfering speakers in the enrollments and the Audio Mixture, respectively.

### 3.2 Speaker Disambiguation via Positive and Negative Enrollments

To show how the Positive and Negative Enrollment pair resolves the ambiguity of which speaker is the target speaker in the noisy enrollments, we introduce the following assumption about speakers' temporal dynamics:

*Assumption*: In a long audio mixture where multiple speakers speak independently, no two speakers will consistently start and stop talking at exactly the same times throughout the mixture.

This assumption is easily satisfied: For two speakers to always talk simultaneously, one speaker would have to deliberately start and stop speaking at the exact same times as the other, effectively aiming to sabotage the other's conversation. Such speakers interfering with intention are unlikely to exist in typical cocktail-party scenario, when interfering speakers in the environment are talking independently from the target speaker. We quantitatively verify this assumption on the real-world audio dataset in Appendix A.

Based on the assumption above, the Positive and Negative Enrollments divide interfering speakers into 4 categories, depending on their existence in both enrollments:

1) Negative Interferer (NI): These are speakers who talk consistently throughout the Positive Enrollment and are present in the Negative Enrollment. The model should learn their voice characteristics from the Negative Enrollment and exclude them from the encoding.

2) Positive Interferer (PI): These are speakers who are not present in the Negative Enrollment and talk in a fraction of the Positive Enrollment. The model should identify those segments when PIs are absent in the Positive Enrollment and exclude these speakers in the encoding. We explore how sensitive our model is to this fractional overlap in Section 4.4.

3) Hybrid Interferer (HI): These speakers appear partially in both the Positive and Negative Enrollments. They resemble a combination of Positive and Negative Interferers, providing the model with two potential cues for suppression: their absence in some Positive Enrollment segments and their presence in some of the Negative Enrollment segments.

4) Neglect-Required Interferer (NRI): These speakers appear exclusively in the Negative Enrollment. Similar to the Negative Interferers, the model encoder should avoid include these speakers' voice characteristic in the extracted embedding.

Since Positive and Negative Interferers represent the two primary scenarios in which the model must learn to distinguish target speakers from interfering ones, we focus our training and evaluation on these two types. We demonstrate our model's generalizability when Hybrid and Neglect-Required Interferers exist in Appendix H and I, respectively.

### 3.3 Model Architecture

As shown in Figure 2, we adopt TF-GridNet [27] as the backbone for both our **Encoding** and **Extraction Branches**. The TF-GridNet architecture consists of a 2D convolution layer followed by stacks of three TF-GridNet blocks. Each block consists of two BiLSTM modules that capture the inter-frequency and temporal information, followed by a Full-band Self-attention Module designed to capture the long-range frame information between frames. To adapt the original TF-GridNet, which

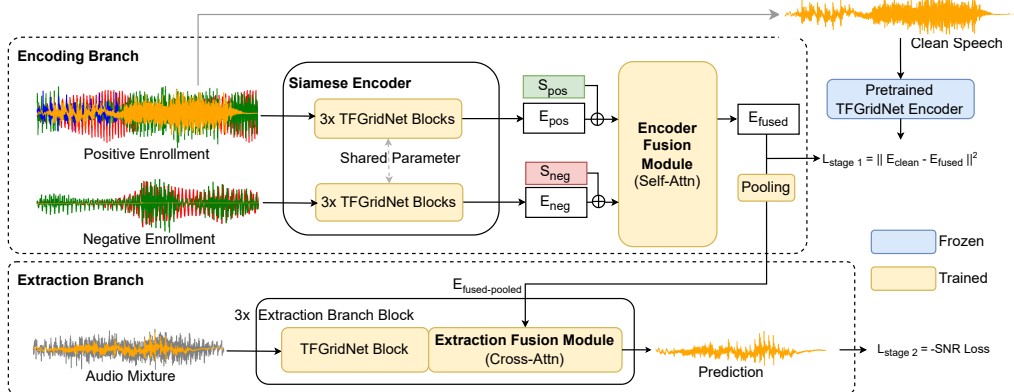

Figure 2: Encoding and Extraction Branch model architecture and training pipeline.

is designed to perform sound separation, to the Target Speaker Extraction task, we introduce an attention-based **Encoder Fusion Module** and an **Extraction Fusion Module**. The Encoder Fusion Module performs comparison between Positive and Negative Enrollment embeddings to generate the embedding of the target speaker, while the Extraction Fusion Module integrates enrollment encoding into the extraction model. The model architecture and training method for encoding the target speaker's feature are shown in Figure 2.

The Positive and Negative Enrollments are first encoded by a pair of TF-GridNet encoders in the **Siamese Encoder**. Since The goal of both branches in the Siamese Encoder is to encode multiple speakers' characteristics from the noisy Positive or Negative Enrollments, we share the parameters between the two TF-GridNet encoders of the Siamese Encoder to reduce model parameter size. The Siamese encoder results in two sequences of embeddings $E_{pos}$ and $E_{neg}$ with shapes $[T_{pos}, D]$ and $[T_{neg}, D]$, respectively. $T_{pos}, T_{neg}$ are the numbers of frames in the positive and negative enrollment embeddings, and $D$ is the feature dimension at each frame.

**The Encoder Fusion Module** then extracts the target speaker's voice embedding from the Siamese Encoders' output $E_{pos}$ and $E_{neg}$. The pseudo-code for the Encoder Fusion Module is shown in Figure 3. In Step 1, two segment embeddings, $S_{pos}$ and $S_{neg}$ with shape $[1, D]$, are first element-wise added to the input embeddings to allow the model to distinguish which enrollment each embedding originates from. The resulting two embeddings are concatenated along the temporal dimension in Step 2, and passed through two Full-band Self-attention layers in Step 3. In these two layers, the self-attention calculation between the embeddings of different Positive Enrollment frames allows the model to identify the Positive Interferers, who remain silent in some of the Positive Enrollment frames, and exclude these speakers in the extracted embedding. Similarly, attention between the Positive and Negative Enrollments embeddings enables the model to identify and exclude Negative Interferers, (i.e., speakers present only in the negative enrollments), from the output embedding. Finally, in Step 4, we truncate the output to retain only the embeddings corresponding to the positive

```
# Input:
#   E_pos: shape [T_pos, D]
#   E_neg: shape [T_neg, D]

# Learnable parameters:
#   Segment Embeddings:
#     S_pos, S_neg: shape [1, D]
#   M: 2 Full-band Self-attention layers

Encoder Fusion Module:
  # Step 1: Elementwise add via broadcasting
  E_pos = E_pos + S_pos # shape [T_pos, D]
  E_neg = E_neg + S_neg # shape [T_neg, D]

  # Step 2: Concatenate along the temporal dimension
  E_concat = [E_pos, E_neg] # shape [T_pos + T_neg, D]

  # Step 3: Apply two Full-band Self-attention layers
  E_concat = M(E_concat) # shape [T_pos + T_neg, D]

  # Step 4: Truncate embeddings
  E_fused = E_concat[:T_pos] # shape [T_pos, D]
```

Figure 3: Encoder Fusion Module pseudo-code.

enrollment as the extracted target speaker embedding, which reduces the output to a $[T_{pos}, D]$ shaped embedding $E_{fused}$. This reduction in embedding size ensures the embedding feature is the same size as the pretrained encoder's output, allowing the model to benefit from the fast convergence brought by training the encoder using knowledge distillation (as shown in Section 4.3). To further reduce the computation load in the extraction branch, we apply non-overlapping average pooling with 40 kernel size along the temporal dimension to $E_{fused}$, resulting in a $[T_{pos}//40, D]$ shaped feature $E_{fused-pooled}$. We discuss the effect of different kernel sizes on the model performance and inference time in Appendix D.

Conditioned on the pooled enrollment encoder output $E_\text{fused-pooled}$, we use a TF-GridNet-based **Extraction Branch** to extract the target speaker's voice from the Audio Mixture. To perform causal target speaker extraction, we modify the TF-GridNet block by adopting the same modification as LookOnceToHear [16]. Three causal TF-GridNet blocks are used in the Extraction Branch. Two cross-attention-based **Extraction Fusion Blocks** are added after the first two causal TF-GridNet blocks to integrate the pooled target speaker embedding $E_\text{fused-pooled}$ into the extraction model. In each fusion module, the pooled target speaker embeddings $E_\text{fused-pooled}$ serve as the Key and Value, while the output from the previous TF-GridNet block serves as the Query for the fusion module. The output from the last TF-GridNet block passes through a transposed convolution layer and an ISTFT module to generate the speech waveform.

In addition to the monaural model described above, which processes single-channel audio, we propose a binaural variant designed to encode and extract binaural audios. The difference between the monaural and binaural model architectures lies solely in the input. The binaural model takes in 4 channels input (stacked real and imaginary components from both channels) and the monaural model takes in only 2 channels (stacked real and imaginary component). By stacking the two channels' STFT, the binaural model can capture the inter-channel temporal differences between the two channels, which encode the directional information of the target speaker since the target speaker speaks at 90 degree azimuth angle in the enrollment. Both model architectures do not differ after the first convolution layer.

### 3.4 Training Method

Due to the additional speaker and background noise in the enrollments, the noisy Positive and Negative Enrollment pairs introduce significant variability in the training data in comparison to the clean enrollments. As shown in Section 4.3, when the Encoding and the Extraction branches are trained together using these noisy training samples, the model converges much more slowly and reaches worse final performance. Motivated by the prior works [16, 28, 29] that successfully trained extraction models conditioned on the clean target speaker embeddings, we separately train the Encoding and Extraction branches in a two-stage training strategy.

As shown in Figure 2, the first stage trains the Siamese Encoder and the Encoder Fusion Module from scratch, using knowledge distillation, to extract the target speaker's embedding from the noisy positive and negative audio enrollments. We obtain the ground-truth target speaker's embedding by passing the clean target speaker's voice through a trained frozen TF-GridNet encoder [16]. Formally, the first stage training loss is written as

$$L_\text{stage 1} = \left\| E_\text{clean} - E_\text{fused} \right\|^2, \tag{2}$$

where $E_\text{clean}$ is the embedding of the clean target speaker voice $a_\text{clean}$ encoded by a trained TF-GridNet encoder from [16].

In the second stage, we train the Extraction Branch to extract the target speaker from the Audio Mixture. The training loss for the second stage is the negative SNR value of the extracted audio $\hat{a}_{tgt}$ and the ground-truth target speaker speech $a_{tgt}$:

$$L_\text{stage 2} = -\text{SNR}(\hat{a}_{tgt}, a_{tgt}). \tag{3}$$

More training and model details are discussed in Appendix C.

## 4 Experiment

### 4.1 Datasets and Baselines

**Datasets** We construct our Positive, Negative, and Mixed Audios from the samples in the Lib- riSpeech dataset [30], with background noise $n^{\{M,P,N\}}$ from the WHAM! dataset [31]. Following the experiment configuration in LookOnceToHear [16], we also construct binaural Positive / Negative / Mixed Audio samples by convolving the speech of each speaker with the binaural RIR data from 4 different datasets: CIPIC [32], RRBRIR [33], ASH-Listening-Set [34], and CATTRIR [35].

Table 1: Comparison between our method and the baselines' performance when extracting from an audio mixture of 2-3 speakers conditioned on an enrollment mixture of 2-4 speakers. NMF's performance does not change as the number of speakers in the Enrollments changes, as it performs unconditional sound separation. Differences in speaker similarity across combinations substantially impact extraction difficulty, leading to the high standard deviation across samples.

| Method | Audio Type | Metric | 2 Speakers in Mixture | | | 3 Speakers in Mixture | | |
|---|---|---|---|---|---|---|---|---|
| | | | 2 Speakers in Enroll | 3 Speakers in Enroll | 4 Speakers in Enroll | 2 Speakers in Enroll | 3 Speakers in Enroll | 4 Speakers in Enroll |
| NMF [36] | Mono | SNRi | 4.24±1.60 | | | 5.65±1.85 | | |
| | | SI-SNRi | -1.65±3.78 | | | -1.50±3.57 | | |
| | | PESQ | 1.05±0.32 | ,, | ,, | 1.22±0.42 | ,, | ,, |
| | | STOI | 0.362±0.139 | | | 0.296±0.114 | | |
| | | DNSMOS | 1.56±0.44 | | | 1.55±0.41 | | |
| | | WER | 0.98±0.05 | | | 0.99±0.02 | | |
| USEF-TFGridnet [17] | Mono | SNRi | 3.42 ± 3.43 | 3.45 ± 3.58 | 3.30 ± 3.50 | 4.31 ± 2.52 | 4.15 ± 2.47 | 4.23 ± 2.58 |
| | | SI-SNRi | -0.03 ± 5.97 | -0.03 ± 6.42 | -0.14 ± 6.00 | 0.29 ± 3.38 | 0.11 ± 3.36 | 0.09 ± 3.24 |
| | | PESQ | 1.52 ± 0.49 | 1.54 ± 0.49 | 1.52 ± 0.49 | 1.32 ± 0.38 | 1.32 ± 0.37 | 1.31 ± 0.36 |
| | | STOI | 0.43 ± 0.17 | 0.43 ± 0.18 | 0.43 ± 0.17 | 0.36 ± 0.11 | 0.35 ± 0.11 | 0.36 ± 0.11 |
| | | DNSMOS | 1.37 ± 0.45 | 1.37 ± 0.45 | 1.36 ± 0.45 | 1.35 ± 0.41 | 1.33 ± 0.41 | 1.34 ± 0.42 |
| | | WER | 0.66 ± 0.33 | 0.66 ± 0.36 | 0.68 ± 0.32 | 0.85 ± 0.26 | 0.86 ± 0.21 | 0.86 ± 0.20 |
| TCE [15] | Mono | SNRi | 8.48±2.34 | 7.84±2.56 | 7.52±2.68 | 8.47±2.37 | 8.57±2.34 | 8.07±2.40 |
| | | SI-SNRi | 6.67±3.69 | 5.57±4.47 | 4.77±5.34 | 6.63±3.74 | 5.13±4.07 | 4.10±4.60 |
| | | PESQ | 1.91 ± 0.34 | 1.80 ± 0.41 | 1.73 ± 0.46 | **1.91 ± 0.34** | 1.53 ± 0.39 | 1.48 ± 0.40 |
| | | STOI | 0.682±0.120 | 0.650±0.140 | 0.624±0.160 | **0.682±0.120** | 0.551±0.143 | 0.521±0.151 |
| | | DNSMOS | 1.85±0.44 | 1.84±0.44 | 1.83±0.44 | 1.85±0.44 | 1.65±0.41 | 1.65±0.41 |
| | | WER | 0.73 ± 0.16 | 0.76 ± 0.24 | 0.77 ± 0.24 | 0.73 ± 0.25 | 0.88 ± 0.21 | 0.88 ± 0.17 |
| Ours (Film fusion) | Mono | SNRi | 8.53±2.30 | 8.39±2.37 | 8.37±2.50 | 9.03±2.23 | 9.00±2.31 | 8.89±2.38 |
| | | SI-SNRi | 6.70±3.43 | 6.45±3.74 | 6.31±3.84 | 6.25±3.35 | 6.06±3.68 | 5.76±3.95 |
| | | PESQ | 1.84 ± 0.32 | 1.83 ± 0.34 | 1.84 ± 0.33 | 1.56 ± 0.32 | 1.54 ± 0.35 | 1.54 ± 0.35 |
| | | STOI | 0.681±0.111 | 0.675±0.123 | 0.665±0.127 | 0.581±0.123 | 0.578±0.129 | 0.565±0.133 |
| | | DNSMOS | 1.92±0.39 | 1.90±0.39 | 1.90±0.41 | 1.69±0.38 | 1.68±0.38 | 1.67±0.38 |
| | | WER | 0.54 ± 0.37 | 0.54 ± 0.29 | 0.54 ± 0.28 | 0.73 ± 0.24 | 0.72 ± 0.25 | 0.73 ± 0.24 |
| Ours (Monaural) | Mono | SNRi | **10.14±2.57** | **9.93±2.71** | **9.79±2.86** | **10.42±2.49** | **10.37±2.64** | **10.22±2.79** |
| | | SI-SNRi | **8.85±3.67** | **8.54±4.01** | **8.30±4.47** | **8.42±3.62** | **8.29±4.06** | **7.82±4.62** |
| | | PESQ | **2.07 ± 0.34** | **2.06 ± 0.36** | **2.05 ± 0.36** | **1.79 ± 0.33** | **1.78 ± 0.34** | **1.75 ± 0.37** |
| | | STOI | **0.758±0.107** | **0.749±0.121** | **0.742±0.126** | **0.668±0.123** | **0.665±0.130** | **0.648±0.146** |
| | | DNSMOS | **2.14±0.37** | **2.13±0.37** | **2.02±0.38** | **1.93±0.37** | **1.92±0.38** | **1.90±0.38** |
| | | WER | **0.42 ± 0.35** | **0.43 ± 0.28** | **0.45 ± 0.28** | **0.61 ± 0.28** | **0.61 ± 0.28** | **0.62 ± 0.28** |

In evaluation, 5000 samples are generated from the LibriSpeech dataset *test-clean* component using the same strategy as the training samples. We report the average improvement in SNR (SNRi) and SI-SNR (SI-SNRi), and the extracted audio's STOI and DNSMOS, along with the standard deviation across evaluation samples calculated using the built-in Pytorch function under the assumption that model performance metrics follow a normal distribution. We detail the training and testing data generation in Appendix B.

**Baselines** We compare our monaural model's performance with TCE [15], USEF-TFGridnet [17], and non-negative matrix factorization (NMF) method [36]. We compare our binaural model's reverberant target speech extraction performance with the LookOnceToHear model [16].

1. TCE [15] considers speakers whose voices do not overlap with a given speaker as the target speakers and extracts their voices. The model takes in a d-vector embedding of the given speaker and removes all interfering speakers who talk at the same time as the given speaker.

2. LookOnceToHear [16] is a binaural target speech extraction method conditioned on noisy audio enrollments. This method uses beamforming to extract the target speaker's characteristics from the noisy Positive Enrollment, which contains audio from multiple additional speakers.

3. USEF-TFGridnet [17] is a target speech extraction model using clean target speaker enrollment as the extraction condition. We include this model as a baseline to show the influence of having interfering speakers in the audio enrollment on the model performance.

4. NMF [36] is a non-deep-learning-based audio separation technique. Since the method could not perform target speaker extraction conditioned on noisy audio examples, we report its extraction quality by selecting the audio with the highest SNR among its output as its extracted audio.

## 4.2 Result on Monaural and Binaural Reverberant Target Speech Extraction

We compare our model with baseline methods under scenarios where different numbers of speakers are present. Table 1 shows the monaural target speaker extraction performance. USEF-TFGridnet [17] shows significantly worse performance, especially as the number of interfering speaker increase

Table 2: Comparison between our method and the baselines' performance when extracting binaural audio from mixtures of 2-3 speakers conditioned on an enrollment mixture of 2-4 speakers.

| Method | Audio Type | Metric | 2 Speakers in Mixture | | | 3 Speakers in Mixture | | |
| --- | --- | --- | --- | --- | --- | --- | --- | --- |
| | | | 2 Speakers in Enroll | 3 Speakers in Enroll | 4 Speakers in Enroll | 2 Speakers in Enroll | 3 Speakers in Enroll | 4 Speakers in Enroll |
| LookOnceToHear [16] | Binaural | SNRi | 9.36±3.56 | 8.90±3.57 | 9.12±3.57 | 9.30±3.39 | 9.28±3.49 | 9.30±3.56 |
| | | SI-SNRi | 7.75±5.35 | 7.24±5.43 | 7.53±5.35 | 6.58±5.37 | 6.49±5.62 | 6.51±5.64 |
| | | PESQ | **2.28 ± 0.47** | **2.25 ± 0.52** | **2.28 ± 0.45** | **1.87 ± 0.48** | **1.88 ± 0.44** | **1.90 ± 0.47** |
| | | STOI | **0.736±0.152** | 0.723±0.162 | **0.732±0.153** | **0.620±0.175** | **0.610±0.180** | **0.611±0.187** |
| | | DNSMOS | 1.79±0.56 | 1.77±0.55 | 1.76±0.55 | **1.59±0.47** | **1.60±0.48** | 1.58±0.49 |
| | | WER | **0.45 ± 0.41** | **0.44 ± 0.35** | 0.47 ± 0.68 | 0.66 ± 0.36 | **0.63 ± 0.33** | **0.64 ± 0.33** |
| Ours (Binaural) | Binaural | SNRi | **9.84±3.57** | **9.60±3.57** | **9.59±3.63** | **9.81±3.28** | **9.78±3.23** | **9.83±3.30** |
| | | SI-SNRi | **8.18±4.81** | **7.84±5.20** | **7.75±5.17** | **6.76±5.27** | **6.72±5.08** | **6.71±5.50** |
| | | PESQ | 2.24± 0.47 | 2.22 ± 0.47 | 2.25 ± 0.47 | 1.85 ± 0.44 | 1.85 ± 0.42 | 1.87 ± 0.43 |
| | | STOI | 0.735±0.150 | **0.729±0.155** | 0.720±0.159 | 0.608±0.177 | 0.605±0.177 | 0.605±0.182 |
| | | DNSMOS | **1.82±0.60** | **1.80±0.59** | **1.80±0.59** | **1.59±0.50** | **1.60±0.49** | **1.59±0.48** |
| | | WER | **0.45 ± 0.48** | 0.45 ± 0.34 | **0.44 ± 0.35** | **0.63 ± 0.33** | 0.64 ± 0.43 | **0.64 ± 0.42** |

in the enrollment. TCE [15] achieves better performance than USEF-TFGridnet [17] baseline since it does not rely on the clean target speaker's enrollment. However, apart from the STOI metric in one scenario, the TCE model shows worse performance than our method under all different numbers of interfering speakers in the Audio Mixture and in enrollments.

In the multi-channel target speech extraction, prior work [16] has explored using beamforming to extract the target speaker's characteristics. In this experiment, we show that additionally including the Negative Enrollment helps improve model performance. As shown in Table 2, in comparison to the LookOnceToHear [16] baseline, based on two-sample t-tests over the 5000 test samples at the 90% confidence level, our model achieves statistically significant improvements in SNRi and SI-SNRi under all conditions. The worse STOI performance might be caused by the difference in parameter size, as shown in Appendix C. In contrast to the monaural experiment results, the binaural models show lower performance in most of the metrics and scenarios. This performance drop might be caused by the difference in training objective, since the binaural models need to predict the reverberant binaural audio, while the monaural models are trained to extract non-reverberant speech (to achieve fair comparison with the prior works focusing on non-reverberant extraction). The DNSMOS, PESQ, and WER metric are also affected by the reverberation effect in the prediction, thus providing less effective information for comparing model performance. We include them here only for completeness.

### 4.3 Ablation Study

**Effectiveness of the two-stage model training** In this section, we show the effectiveness of the two-stage training by showing the validation loss curve of our model with and without the first training stage. As shown in Figure 4, end-to-end training reaches 3 dB SNR on the validation set after 600k optimization steps (around 125 hours), while the combined training time for the two-staged training takes 240k optimization steps (around 50 hours) based on a single Nvidia A10 24GB GPU.

Such a performance difference might be caused by the high difficulty in encoding the expected target speaker from the noisy enrollment pair. Since its very challenging for the model to draw similarity between the ground-truth single speaker audio and the noisy pair of enrollments input, training using an end-to-end learning method provides little guidance to the encoder on which speaker is expected

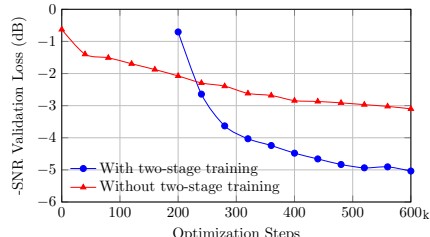

Figure 4: Validation loss values for the optimization step. The curve for the two-stage training begins at the 200k step to account for the 200k optimization steps performed during the first training stage.

to be encoded. In contrast, the two-stage training explicitly guides the model to extract the target speaker's embedding in the first stage, significantly accelerating convergence.

**Cross-Attention based Fusion over Film Fusion Method** Film fusion is widely applied in the prior target audio extraction works [37, 15, 16]. However, Film fusion passes the condition embedding

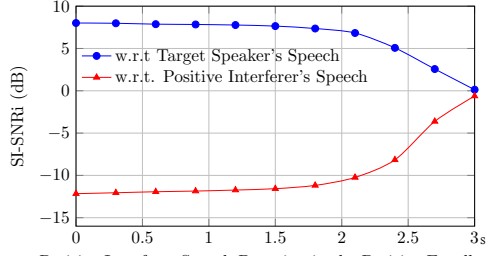
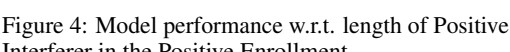
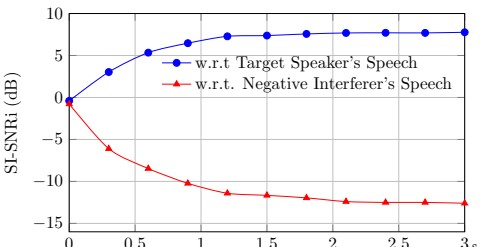

Figure 4: Model performance w.r.t. length of Positive Interferer in the Positive Enrollment.

Figure 5: Model performance w.r.t. length of Negative Interferer in the Negative Enrollment.

Table 3: SI-SDRi of 10 Sample using Ground-Truth or Inaccurate User Enrollment Labelings.

| Enrollment Labeling Source | 1 | 2 | 3 | 4 | 5 | 6 | 7 | 8 | 9 | 10 |
|---|---|---|---|---|---|---|---|---|---|---|
| Ground Truth | 10.98 | 7.60 | 8.35 | 11.71 | 10.31 | 12.31 | 11.01 | 7.03 | 8.00 | 11.81 |
| User-Labelings | 10.90 ± 0.04 | 7.60 ±0.05 | 8.40 ±0.02 | 11.65 ±0.04 | 10.23 ±0.07 | 12.14 ±0.12 | 10.82 ±0.24 | 7.32 ±1.32 | 8.04 ±0.10 | 11.15 ±2.10 |

through a linear layer, and element-wise multiplies the output with the input tensor to perform fusion. This limits the model to use a fixed embedding size, which might not be capable of encoding the fine-grained details of the target speaker's voice characteristics. In this experiment, we show that our attention-based fusion method is more optimal for our target speech extraction task.

We keep the rest of the model architecture intact and only change the model fusion method. In particular, we perform global average pooling on the encoder head output along the temporal dimension to obtain an embedding as the condition input to the Film fusion block. As shown in Table 1, our attention-based fusion method achieves higher performance in all application scenarios. This shows that the cross-attention-based fusion method is more preferable for the target speech extraction conditioned on noisy enrollments.

## 4.4 Model Performance in Challenging Application Scenarios

**Model Performance under different PI and NI's speech length** As the Positive Interferer (PI)'s speech duration in the Positive Enrollment increases, fewer frames are available for the model to distinguish the Target Speaker from the PI. Similarly, reducing the Negative Interferer (NI)'s speech length in the Negative Enrollment results in less available conditional information to remove NI's voice characteristics in the target speaker encoding. In this section, we explicitly examine how the model performance varies with the duration of the PI and NI's speech in the enrollment.

We focus on the scenario where one PI and one NI exist in the enrollments, and the Audio Mixture contains the same interferer as those in the enrollments. Let the length of the PI's speech in the Positive Enrollment be $l_{pos}$, and the NI's speech in the Negative Enrollment be $l_{neg}$. In Figure 4, we report the average model performance when extracting 2000 test samples with $l_{neg}$ randomly sampled between 1s to 3s and $l_{pos}$ takes different values between 0s to 3s. We evaluate the extracted audio's SI-SNRi with respect to the target speaker and the PI's speech in the Audio Mixture. When $l_{pos} = 3$, the model fails to distinguish the two speakers as expected, since both speakers have the same speech length in each enrollment. As $l_{pos}$ decreases below 2.7s, the model correctly identifies the target speaker's characteristic, as shown by the increasing SI-SNRi of the extracted audio w.r.t. the target speaker's speech and the decreasing SI-SNRi w.r.t. the PI's speech. Alternatively, we randomly sample $l_{pos}$ between 1s to 3s and evaluate the model performance when $l_{neg}$ takes a value between 0s to 3s. As shown in Figure 5, the model correctly extracts the target speaker and removes the NI's speech when $l_{neg}$ increases above 0.3s. These experiments demonstrate that the model indeed leverages the differences in speakers' speech length in both enrollments to encode the target speaker, and could distinguish speakers from as short as 0.3 seconds of misalignment between their speech.

**Model performance under inaccurate Positive and Negative Labeling** In the real-world application scenario, inaccurate Positive and Negative Enrollment labeling from the users could lead to

unwanted inclusion of the target speaker's voice in the Negative Enrollments, or including timesteps when the target speaker remains silent in the Positive Enrollment. Both cases violate our model input assumption. In this section, we demonstrate our model's tolerance to the inaccuracy in the user-provided Enrollments' labeling.

We randomly select 2-3 speakers and overlay their voices so that each speaker spans 5-7 seconds of a 10-second audio mixture. For 10 such randomly generated samples, we instruct 10 users to label the start and stop timesteps of a certain speaker following a speaker hint, e.g, the same speaker heard in another clean audio example. The segments not labeled by the user are considered negative enrollments. Using the enrollment labeling provided by users, we perform extraction on another audio mixture containing the target speaker. We report the SI-SNRi of each sample across different users' labeling, as well as the SI-SNRi if the ground truth labeling is used. As shown in Table 3, our model shows little performance degradation when using the inaccurate positive enrollment labeling provided by the user, despite each start and stop timestep of the positive enrollment being on average $0.30 \pm 0.19$ seconds away from the ground truth timestep. In addition, the model shows a small variation in metric value across different users' labeling. This validates our model's robustness to the inaccuracy in different users' labeling.

**Model Performance on Real-World Audio Mixtures**

In this section, we compare our model and the baseline methods on naturally recorded real-world audio mixtures. We evaluate the Mean Opinion Score (MOS) of different models using five naturally recorded audio mixtures sourced from Freesound.org and the VoxConverse dataset. These sounds are noisy real-world speech recordings captured in pubs, metro stations, urban areas, and city council meetings. For the evaluation, we manually labelled the Positive and Negative Enrollment and extracted target speech from Audio Mixture clips taken from the same recording. The duration of the enrollment and Audio Mixture clips range from 3 to 8 seconds.

Table 4: SI-SDRi under Inaccurate User labelings.

| Method | MOS |
| --- | --- |
| Ours (Monaural) | 3.35 |
| Ours (FiLM Fusion) | 2.60 |
| USEF-TFGridnet | 1.45 |
| Unprocessed | 2.10 |

10 participants are asked to rate the original (unprocessed) Audio Mixture, and the audio extracted by three models. Given a textual description of the target speaker (e.g., the female speaker talking over the crowd), they rated the clarity of the target speech relative to background noise on a 1–4 scale. As shown in Table 4, our model achieved a MOS of 3.35, outperforming all baseline methods. These results demonstrate the superiority and practical applicability of our model in real-world target speech extraction scenarios.

## 5    Conclusion and Limitation

In this paper, we present a method for performing target speech extraction conditioned on noisy enrollments where interfering speakers' speech overlaps significantly with the target speaker. We optimize our model to encode the target speaker from easily obtained noisy positive and negative enrollments, without using any clean audio enrollment input. Experiments show that our method achieves SOTA performance under the challenging and realistic application scenario.

Though the proposed two-stage training strategy for the encoding branch significantly accelerates convergence, it may limit the encoder's performance to that of the frozen encoder used for knowledge distillation. Further work on encoding branch training is required to remove such potential performance limitations while maintaining the high convergence speed. In addition, our model extraction still includes artifacts noticeable to the human ear, as shown by the low DNSMOS score reported. This hinders the model's applicability to downstream tasks such as audio recognition. Further work is required to reduce the artifacts, for example, by employing more advanced extraction branch architectures. Finally, although our model can flexibly perform extraction using clean or noisy audio enrollments, it does not outperform SOTA models explicitly trained to extract conditioned on clean enrollments. Further work is required to develop models with strong generalizability across both enrollment strategies.

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

# NeurIPS Paper Checklist

1. **Claims**

   Question: Do the main claims made in the abstract and introduction accurately reflect the paper's contributions and scope?

   Answer: [Yes]

   The abstract and introduction clearly describe the main contributions of the paper, including using noisy positive and negative enrollments for target speaker extraction, a novel model architecture, and a two-stage training method. These claims are well supported by the experiments and analysis in the main text, and the paper also discusses assumptions and limitations, matching the stated scope.

   Guidelines:

   - The answer NA means that the abstract and introduction do not include the claims made in the paper.
   - The abstract and/or introduction should clearly state the claims made, including the contributions made in the paper and important assumptions and limitations. A No or NA answer to this question will not be perceived well by the reviewers.
   - The claims made should match theoretical and experimental results, and reflect how much the results can be expected to generalize to other settings.
   - It is fine to include aspirational goals as motivation as long as it is clear that these goals are not attained by the paper.

2. **Limitations**

   Question: Does the paper discuss the limitations of the work performed by the authors?

   Answer: [Yes]

   Justification: The paper discusses several limitations in the main experimental results section and the final "Conclusion and Limitation" section. It acknowledges challenges such as performance ceiling posed by the proposed two-stage training strategy, the gap compared to models trained with clean enrollments, and the difficulty of predicting reverberant audio. It also notes potential issues like artifacts in extracted speech and the need for better encoding architectures. Additionally, the paper reflects on the assumption that speakers do not start and stop speaking in perfect synchrony, and empirically validates this assumption in the "Model Performance in Challenging Application Scenarios" section.

   Guidelines:

   - The answer NA means that the paper has no limitation while the answer No means that the paper has limitations, but those are not discussed in the paper.
   - The authors are encouraged to create a separate "Limitations" section in their paper.
   - The paper should point out any strong assumptions and how robust the results are to violations of these assumptions (e.g., independence assumptions, noiseless settings, model well-specification, asymptotic approximations only holding locally). The authors should reflect on how these assumptions might be violated in practice and what the implications would be.
   - The authors should reflect on the scope of the claims made, e.g., if the approach was only tested on a few datasets or with a few runs. In general, empirical results often depend on implicit assumptions, which should be articulated.
   - The authors should reflect on the factors that influence the performance of the approach. For example, a facial recognition algorithm may perform poorly when image resolution is low or images are taken in low lighting. Or a speech-to-text system might not be used reliably to provide closed captions for online lectures because it fails to handle technical jargon.
   - The authors should discuss the computational efficiency of the proposed algorithms and how they scale with dataset size.
   - If applicable, the authors should discuss possible limitations of their approach to address problems of privacy and fairness.

- While the authors might fear that complete honesty about limitations might be used by reviewers as grounds for rejection, a worse outcome might be that reviewers discover limitations that aren't acknowledged in the paper. The authors should use their best judgment and recognize that individual actions in favor of transparency play an important role in developing norms that preserve the integrity of the community. Reviewers will be specifically instructed to not penalize honesty concerning limitations.

3. **Theory assumptions and proofs**

   Question: For each theoretical result, does the paper provide the full set of assumptions and a complete (and correct) proof?

   Answer: [NA]

   Justification: The paper does not include any formal theoretical results, theorems, or proofs. While it introduces an assumption about speaker temporal dynamics (i.e., that speakers do not start and stop speaking in perfect synchrony), this is empirically validated on the real-world datasets in Appendix A rather than formally proven. Therefore, this question is not applicable.

   Guidelines:

   - The answer NA means that the paper does not include theoretical results.
   - All the theorems, formulas, and proofs in the paper should be numbered and cross-referenced.
   - All assumptions should be clearly stated or referenced in the statement of any theorems.
   - The proofs can either appear in the main paper or the supplemental material, but if they appear in the supplemental material, the authors are encouraged to provide a short proof sketch to provide intuition.
   - Inversely, any informal proof provided in the core of the paper should be complemented by formal proofs provided in appendix or supplemental material.
   - Theorems and Lemmas that the proof relies upon should be properly referenced.

4. **Experimental result reproducibility**

   Question: Does the paper fully disclose all the information needed to reproduce the main experimental results of the paper to the extent that it affects the main claims and/or conclusions of the paper (regardless of whether the code and data are provided or not)?

   Answer: [Yes]

   Justification: The paper provides sufficient details to reproduce the main experimental results. It describes the dataset construction process using LibriSpeech and WHAM!, outlines the model architecture clearly, and explains the training procedure, including the two-stage training strategy. Hyperparameters, evaluation metrics, and comparison baselines are also specified. Additionally, the paper mentions that code, model checkpoints, pretrained models, and usage instructions (readme file) are included in the supplementary material, supporting reproducibility.

   Guidelines:

   - The answer NA means that the paper does not include experiments.
   - If the paper includes experiments, a No answer to this question will not be perceived well by the reviewers: Making the paper reproducible is important, regardless of whether the code and data are provided or not.
   - If the contribution is a dataset and/or model, the authors should describe the steps taken to make their results reproducible or verifiable.
   - Depending on the contribution, reproducibility can be accomplished in various ways. For example, if the contribution is a novel architecture, describing the architecture fully might suffice, or if the contribution is a specific model and empirical evaluation, it may be necessary to either make it possible for others to replicate the model with the same dataset, or provide access to the model. In general. releasing code and data is often one good way to accomplish this, but reproducibility can also be provided via detailed instructions for how to replicate the results, access to a hosted model (e.g., in the case of a large language model), releasing of a model checkpoint, or other means that are appropriate to the research performed.

- While NeurIPS does not require releasing code, the conference does require all submissions to provide some reasonable avenue for reproducibility, which may depend on the nature of the contribution. For example
  (a) If the contribution is primarily a new algorithm, the paper should make it clear how to reproduce that algorithm.
  (b) If the contribution is primarily a new model architecture, the paper should describe the architecture clearly and fully.
  (c) If the contribution is a new model (e.g., a large language model), then there should either be a way to access this model for reproducing the results or a way to reproduce the model (e.g., with an open-source dataset or instructions for how to construct the dataset).
  (d) We recognize that reproducibility may be tricky in some cases, in which case authors are welcome to describe the particular way they provide for reproducibility. In the case of closed-source models, it may be that access to the model is limited in some way (e.g., to registered users), but it should be possible for other researchers to have some path to reproducing or verifying the results.

5. **Open access to data and code**

   Question: Does the paper provide open access to the data and code, with sufficient instructions to faithfully reproduce the main experimental results, as described in supplemental material?

   Answer: [Yes]

   Justification: The code, model checkpoints, and the instructions for reproducing the result are given in the supplementary file.

   Guidelines:
   - The answer NA means that paper does not include experiments requiring code.
   - Please see the NeurIPS code and data submission guidelines (`https://nips.cc/public/guides/CodeSubmissionPolicy`) for more details.
   - While we encourage the release of code and data, we understand that this might not be possible, so "No" is an acceptable answer. Papers cannot be rejected simply for not including code, unless this is central to the contribution (e.g., for a new open-source benchmark).
   - The instructions should contain the exact command and environment needed to run to reproduce the results. See the NeurIPS code and data submission guidelines (`https://nips.cc/public/guides/CodeSubmissionPolicy`) for more details.
   - The authors should provide instructions on data access and preparation, including how to access the raw data, preprocessed data, intermediate data, and generated data, etc.
   - The authors should provide scripts to reproduce all experimental results for the new proposed method and baselines. If only a subset of experiments are reproducible, they should state which ones are omitted from the script and why.
   - At submission time, to preserve anonymity, the authors should release anonymized versions (if applicable).
   - Providing as much information as possible in supplemental material (appended to the paper) is recommended, but including URLs to data and code is permitted.

6. **Experimental setting/details**

   Question: Does the paper specify all the training and test details (e.g., data splits, hyperparameters, how they were chosen, type of optimizer, etc.) necessary to understand the results?

   Answer: [Yes]

   Justification: The paper provides detailed descriptions of the training and testing setup. It specifies the datasets used, the construction of positive, negative, and mixed audio samples, and the baseline models for comparison. Also, some specifics (e.g., hyperparameters, optimizer details) are deferred to the appendix B, C. We use the official train/validation/test of the LibriSpeech dataset to construct different dataset splits.

   Guidelines:

- The answer NA means that the paper does not include experiments.
- The experimental setting should be presented in the core of the paper to a level of detail that is necessary to appreciate the results and make sense of them.
- The full details can be provided either with the code, in appendix, or as supplemental material.

7. **Experiment statistical significance**

   Question: Does the paper report error bars suitably and correctly defined or other appropriate information about the statistical significance of the experiments?

   Answer: [Yes]

   Justification: The paper reports standard deviations alongside the main evaluation metrics in the result tables, indicating variability across test samples. It also states that the metrics are assumed to follow a normal distribution and that the standard deviations are computed.

   Guidelines:
   - The answer NA means that the paper does not include experiments.
   - The authors should answer "Yes" if the results are accompanied by error bars, confidence intervals, or statistical significance tests, at least for the experiments that support the main claims of the paper.
   - The factors of variability that the error bars are capturing should be clearly stated (for example, train/test split, initialization, random drawing of some parameter, or overall run with given experimental conditions).
   - The method for calculating the error bars should be explained (closed form formula, call to a library function, bootstrap, etc.)
   - The assumptions made should be given (e.g., Normally distributed errors).
   - It should be clear whether the error bar is the standard deviation or the standard error of the mean.
   - It is OK to report 1-sigma error bars, but one should state it. The authors should preferably report a 2-sigma error bar than state that they have a 96% CI, if the hypothesis of Normality of errors is not verified.
   - For asymmetric distributions, the authors should be careful not to show in tables or figures symmetric error bars that would yield results that are out of range (e.g. negative error rates).
   - If error bars are reported in tables or plots, The authors should explain in the text how they were calculated and reference the corresponding figures or tables in the text.

8. **Experiments compute resources**

   Question: For each experiment, does the paper provide sufficient information on the computer resources (type of compute workers, memory, time of execution) needed to reproduce the experiments?

   Answer: [Yes]

   Justification: The paper specifies that training was conducted using a single Nvidia A10 24GB GPU and provides concrete information about training time and optimization steps in Appendix C. This level of detail is sufficient to estimate the computational requirements for reproducing the experiments.

   Guidelines:
   - The answer NA means that the paper does not include experiments.
   - The paper should indicate the type of compute workers CPU or GPU, internal cluster, or cloud provider, including relevant memory and storage.
   - The paper should provide the amount of compute required for each of the individual experimental runs as well as estimate the total compute.
   - The paper should disclose whether the full research project required more compute than the experiments reported in the paper (e.g., preliminary or failed experiments that didn't make it into the paper).

9. **Code of ethics**

Question: Does the research conducted in the paper conform, in every respect, with the NeurIPS Code of Ethics https://neurips.cc/public/EthicsGuidelines?

Answer: [Yes]

Justification: Our training samples involve only publicly available datasets, with no personal identifiable information. We designed and performed our user study so that it fully complies with the NeurIPS Code of Ethics.

Guidelines:

- The answer NA means that the authors have not reviewed the NeurIPS Code of Ethics.
- If the authors answer No, they should explain the special circumstances that require a deviation from the Code of Ethics.
- The authors should make sure to preserve anonymity (e.g., if there is a special consideration due to laws or regulations in their jurisdiction).

10. **Broader impacts**

Question: Does the paper discuss both potential positive societal impacts and negative societal impacts of the work performed?

Answer: [Yes]

Justification: We discussed the social impact of the work in the main paper. Our paper focuses on improving making it easier for users to focus on desired voices in complex auditory scenes, an application scenario already widely used. No social impact we feel should be explicitly addressed here.

Justification: We discuss both potential positive and negative societal impacts in the Appendix. Our work aims to help users better focus on desired voices in complex auditory scenes, which is an application scenario already widely used. There are many potential societal consequences of our work, none of which we feel must be specifically highlighted here.

Guidelines:

- The answer NA means that there is no societal impact of the work performed.
- If the authors answer NA or No, they should explain why their work has no societal impact or why the paper does not address societal impact.
- Examples of negative societal impacts include potential malicious or unintended uses (e.g., disinformation, generating fake profiles, surveillance), fairness considerations (e.g., deployment of technologies that could make decisions that unfairly impact specific groups), privacy considerations, and security considerations.
- The conference expects that many papers will be foundational research and not tied to particular applications, let alone deployments. However, if there is a direct path to any negative applications, the authors should point it out. For example, it is legitimate to point out that an improvement in the quality of generative models could be used to generate deepfakes for disinformation. On the other hand, it is not needed to point out that a generic algorithm for optimizing neural networks could enable people to train models that generate Deepfakes faster.
- The authors should consider possible harms that could arise when the technology is being used as intended and functioning correctly, harms that could arise when the technology is being used as intended but gives incorrect results, and harms following from (intentional or unintentional) misuse of the technology.
- If there are negative societal impacts, the authors could also discuss possible mitigation strategies (e.g., gated release of models, providing defenses in addition to attacks, mechanisms for monitoring misuse, mechanisms to monitor how a system learns from feedback over time, improving the efficiency and accessibility of ML).

11. **Safeguards**

Question: Does the paper describe safeguards that have been put in place for responsible release of data or models that have a high risk for misuse (e.g., pretrained language models, image generators, or scraped datasets)?

Answer: [NA] .

Justification: The paper does not release models or datasets that are clearly high risk for misuse. It uses publicly available datasets and does not involve sensitive or scraped data.

Guidelines:

- The answer NA means that the paper poses no such risks.
- Released models that have a high risk for misuse or dual-use should be released with necessary safeguards to allow for controlled use of the model, for example by requiring that users adhere to usage guidelines or restrictions to access the model or implementing safety filters.
- Datasets that have been scraped from the Internet could pose safety risks. The authors should describe how they avoided releasing unsafe images.
- We recognize that providing effective safeguards is challenging, and many papers do not require this, but we encourage authors to take this into account and make a best faith effort.

12. **Licenses for existing assets**

Question: Are the creators or original owners of assets (e.g., code, data, models), used in the paper, properly credited and are the license and terms of use explicitly mentioned and properly respected?

Answer: [Yes]

Justification: The paper uses publicly available datasets and properly cites the original sources. It also credits the TF-GridNet model as the architectural backbone with appropriate references. While the paper does not explicitly mention license types or URLs, the cited resources are standard in the community, and their terms of use are widely known. There is no indication of improper use or lack of attribution.

Guidelines:

- The answer NA means that the paper does not use existing assets.
- The authors should cite the original paper that produced the code package or dataset.
- The authors should state which version of the asset is used and, if possible, include a URL.
- The name of the license (e.g., CC-BY 4.0) should be included for each asset.
- For scraped data from a particular source (e.g., website), the copyright and terms of service of that source should be provided.
- If assets are released, the license, copyright information, and terms of use in the package should be provided. For popular datasets, `paperswithcode.com/datasets` has curated licenses for some datasets. Their licensing guide can help determine the license of a dataset.
- For existing datasets that are re-packaged, both the original license and the license of the derived asset (if it has changed) should be provided.
- If this information is not available online, the authors are encouraged to reach out to the asset's creators.

13. **New assets**

Question: Are new assets introduced in the paper well documented and is the documentation provided alongside the assets?

Answer: [Yes]

Justification: The paper introduces a new model and training pipeline for target speaker extraction and provides code, pretrained checkpoints, and instructions in the supplementary material. While it does not create a new dataset, the documentation appears sufficient for users to understand and apply the new assets. The assets are anonymized for submission, aligning with NeurIPS guidelines.

Guidelines:

- The answer NA means that the paper does not release new assets.
- Researchers should communicate the details of the dataset/code/model as part of their submissions via structured templates. This includes details about training, license, limitations, etc.

- The paper should discuss whether and how consent was obtained from people whose asset is used.
- At submission time, remember to anonymize your assets (if applicable). You can either create an anonymized URL or include an anonymized zip file.

14. **Crowdsourcing and research with human subjects**

    Question: For crowdsourcing experiments and research with human subjects, does the paper include the full text of instructions given to participants and screenshots, if applicable, as well as details about compensation (if any)?

    Answer: [Yes]

    Justification: We performed the experiment on 10 more human volunteers to verify the feasibility of obtaining the Positive and Negative Enrollments. The experiment details and the instruction given to the human participants are given in the Appendix.

    Guidelines:

    - The answer NA means that the paper does not involve crowdsourcing nor research with human subjects.
    - Including this information in the supplemental material is fine, but if the main contribution of the paper involves human subjects, then as much detail as possible should be included in the main paper.
    - According to the NeurIPS Code of Ethics, workers involved in data collection, curation, or other labor should be paid at least the minimum wage in the country of the data collector.

15. **Institutional review board (IRB) approvals or equivalent for research with human subjects**

    Question: Does the paper describe potential risks incurred by study participants, whether such risks were disclosed to the subjects, and whether Institutional Review Board (IRB) approvals (or an equivalent approval/review based on the requirements of your country or institution) were obtained?

    Answer: [Yes]

    Justification: We have obtained the approval from the institution where all the authors are from.

    Guidelines:

    - The answer NA means that the paper does not involve crowdsourcing nor research with human subjects.
    - Depending on the country in which research is conducted, IRB approval (or equivalent) may be required for any human subjects research. If you obtained IRB approval, you should clearly state this in the paper.
    - We recognize that the procedures for this may vary significantly between institutions and locations, and we expect authors to adhere to the NeurIPS Code of Ethics and the guidelines for their institution.
    - For initial submissions, do not include any information that would break anonymity (if applicable), such as the institution conducting the review.

16. **Declaration of LLM usage**

    Question: Does the paper describe the usage of LLMs if it is an important, original, or non-standard component of the core methods in this research? Note that if the LLM is used only for writing, editing, or formatting purposes and does not impact the core methodology, scientific rigorousness, or originality of the research, declaration is not required.

    Answer: [NA]

    Justification: LLM is used solely to refine wording for clarity in this work.

    Guidelines:

    - The answer NA means that the core method development in this research does not involve LLMs as any important, original, or non-standard components.
    - Please refer to our LLM policy (`https://neurips.cc/Conferences/2025/LLM`) for what should or should not be described.

The appendix is structured as follow: **Section A** quantitatively verifies the feasibility of encoding the target speaker using the Positive and Negative Enrollments pair. **Section B-D** provide training details and justification for the hyperparameter selection. **Section E-I** present model performance under different numbers and speech lengths of interfering speakers. **Section J-L** reports the model's performance when clean audio enrollments are available. **Section M** show model's generalizability to speech data from other datasets. **Seciton N** shows the successful and failure cases. **Section O** states the societal impact of the work.

**Section A** :Quantitative Evaluation on the Overlapping Ratio between the Target Speaker and the Interfering Speakers

**Section B** :Data Simulation Method

**Section C** :Model Architecture and Training Detail

**Section D** :Embedding Pooling Size

**Section E** :Affect of Increasing Number of Positive and Negative Interferers on Model Performance

**Section F** :Model Performance under Different Lengths of Positive and Negative Enrollment

**Section G** :Model Performance with respect to the Input Audio Quality

**Section H** :Model Performance under presence of Hybrid Interferers

**Section I** :Model Performance under presence of Neglect-Required Interferers

**Section J** :Model Performance for Target speaker Confusion Problem

**Section K** :Model Performance when Clean Target Speaker's Enrollment is available

**Section L** :Model performance on WSJ0 Dataset

**Section M** :Audio Visualization and Failure Cases

**Section N** :Societal Impact

## A   Quantitative Evaluation on the Overlapping Ratio between the Target Speaker and the Interfering Speakers

As shown in Section 3.2, we base our problem formulation on the assumption that speakers will not always start and stop talking at the same time. In this section, we show such temporal misalignment between speakers in real-world noisy audio mixtures is prevalent, which allows our model to identify the target speaker from the noisy Positive and Negative Enrollment.

We validate the assumption using the MSDWild dataset and the VoxConverse dataset. Both datasets are activity detection datasets consisting of real-world conversational audios, and provide ground truth activity labeling indicating which speaker speaks when in each conversation. To investigate the overlapping ratio between speakers from different conversations groups, we perform pairwise comparison between conversation samples from each dataset. Specifically, for a pair of conversation containing speakers $\{s_{a1}, s_{a2}, ..., s_{am}\}$ and $\{s_{b1}, s_{b2}, ..., s_{bn}\}$ respectively, we consider all the speaker pairs from different conversation groups $\{(s_{ai}, s_{bj}) | i \in [1..m], j \in [1..n]\}$, and calculate the mean Intersection Over Union (IOU) for the speech of these speaker pairs $\frac{1}{nm} \sum_{i \in [1..m], j \in [1..n]} IoU(s_{ai}, s_{bj})$.

The mean IOU and the standard deviation of IOU across different samples is $0.25 \pm 0.20$ for the *many-talker val* split of the MSDWild dataset, and $0.22 \pm 0.06$ for the test split of VoxConverse dataset. This indicates that, on average, the non-overlapping speech between any two independent speakers spans more than half of their combined speech segments. These non-overlapped regions offer rich information for our Positive and Negative Enrollment strategy, enabling the system to identify the target speaker and perform accurate speaker extraction. Note that the non-overlapping regions are defined with respect to pairs of speakers. In scenes with multiple interfering speakers, different interferers may overlap with the target speaker at different times, resulting in a mixture where no single timestep contains only one active speaker. However, by obtaining pairwise speaker difference from the non-overlapping speech between two speakers, Positive and Negative Enrollment pair still provide sufficient information to distinguish the target speaker from interfering speakers.

Table 5: Speech length of different types of speakers' speech in enrollments when 3 seconds of Positive and 3 seconds of Negative Enrollments are used.

| Speaker Type | Length in Positive Enrollment | Length in Negative Enrollment |
|---|---|---|
| Target Speaker | 3 seconds | 0 seconds |
| Positive Interferer | 1-2 seconds | 0 seconds |
| Negative Interferer | 3 seconds | 1-3 seconds |
| Hybrid Interferer | 1-2 seconds | 1-3 seconds |
| Neglect-Required Interferer | 0 seconds | 1-3 seconds |

# B    Data Simulation Method

In this section, we detail the method for simulating the training and testing audio.

As shown in Section 3.2 in the main paper, based on the presence/absence of the interfering speaker in all/subset of the segments in the enrollments, the interfering speakers are classified into 4 types. Table 5 summarizes the length of each type of speaker in the Enrollments. **Target Speaker** speaks throughout the Positive Enrollment and does not exist in the Negative Enrollment. **Negative Interferers'** speech fully overlap with the target speaker in the Positive Enrollment and exist in the Negative Enrollment. Alternatively, **Positive Interferers'** speech does not exist in the Negative Enrollment, and exist in a subset of segments in the Positive Enrollment. **Hybrid Interferer** exist in a subset of the Positive Enrollment, and also exist in the Negative Enrollment. **Neglect-Required Interferers** exist only in the Negative Enrollment and are excluded from the Positive Enrollment. In order to ensure each speaker's voice span the desired length of the Enrollment, we use WebRTC Voice Activity Detector [38] to detect and remove zeros in the LibriSpeech dataset samples, before repeating or cutting to construct their voice of desired length in the Audio Mixture, the Positive Enrollment and the Negative Enrollment.

We generate our training, validation, and testing data from the `train-clean-360`, `dev-clean`, and `test-clean` components from the LibriSpeech dataset [30], respectively. The sampling rate is 16000. We generate the training samples so that the enrollments and the Audio Mixture all have one target speaker and two interfering speakers. The target speaker is shared between the enrollments and the Audio Mixture, while the interfering speakers in the Audio Mixtures are two randomly sampled speakers different from the target speaker, without being required to be the same as those in the enrollments. To reduce the instability caused by the variation of different speaker types, the Hybrid and Neglect-Required Interferers are not included in the enrollments in the training samples, so the two interfering speakers are randomly designated as either the Positive Interferer or the Negative Interferer. We enforce the target and interfering speakers to talk throughout the Audio Mixture to increase the extraction difficulty. All training samples are generated on the fly to enhance variability and diversity during training. The testing data generation follow the same strategy as the training samples. All the Audio Mixtures in the test samples are 6 seconds long.

To simulate the background environment noise, we use the WHAM! [31] noise dataset. Different noise samples from the WHAM! noise dataset are recorded in different environments. As a result, additional interfering sounds might exist in one WHAM! noise example but not the other. For instance, some WHAM! noise dataset samples contain music playing in the background while others don't. Using a WHAM! noise sample containing music in the Positive Enrollment and a WHAM! sample of ambient environment noise in the Negative Enrollment will confuse our model on whether the music is the target sound to be extracted. We resolve such ambiguity by sampling the $n^{\{M,P,N\}}$ from the same WHAM! noise sample but at different temporal segments. Selecting random segments from the WHAM! noise prevents the model from assuming the background noise is the same across the Positive, Negative, and Mixed Audio. We scaled the WHAM! noise to be between $[-2.5, 2.5]$ SNR with respect to the ground truth target speaker's voice in each sample.

To simulate the binaural reverberant training and testing samples, we follow the same data simulation method as in the LookOnceToHear [16]. In particular, we convolve each speaker's voice with BRIR from four binaural room impulse response datasets [32, 34, 33, 35]. The binaural RIRs used for constructing one data sample are randomly selected such that 1. All BRIR used in a single data sample generation are from the same scene, and 2. The BRIR for the target speaker in the Positive

Table 6: Comparison of model parameter size, inference time, and memory usage when extracting 1-second audio on an Intel Xeon Silver 4314 @ 2.40GHz CPU. We report the average inference time for extraction in 50 repeated experiments. Replacing the Film fusion module with the cross-attention based fusion module in our final model architecture could significantly reduce the parameter size.

| Model | Param Size | Inference Time | Inference Memory Usage |
|---|---|---|---|
| Ours (cross attn) | 1.88 M | 0.36s | 2.30 GB |
| Ours (Film fusion) | 3.94 M | 0.3s | 2.31 GB |
| TCE | 2.54 M | 0.11s | 1.89 GB |
| LookOnceToHear | 4.41 M | 0.35s | 2.32 GB |

Enrollment has the direction of arrival of 0 degrees. The pre-processing strategy for each speaker's voice and the generation of background noise remains the same as the monaural dataset.

## C  Model Architecture and Training Detail

The TF-GridNet [27] Encoder in the Encoding Branch uses the following configuration: $4 \times 4$ kernel size and $1 \times 1$ stride in the first Conv2D, 64 hidden units in all three BiLSTM layers, 8 attention head numbers in the Full-band Self-attention Module. The input audio is processed by Short-Time Fourier Transform (STFT) with 128 window size and 64 hop length.

To perform causal inference in the extraction branch, we follow the same modification as in the LookOnceToHear [16]. In particular, we remove the global layer normalization after the first convolution layer, change the BiLSTM in the TF-GridNet blocks to unidirectional LSTM, and constrain the Full-band Self-attention Modules to calculate the causal attention value of a one-time frame with only frames before it. The parameter configurations of causal TF-GridNet blocks in the Extraction Branch are the same as the Encoding Branch, apart from using $1 \times 1$ kernel size in its first Conv2D layer. Three causal TF-GridNet blocks are used in the Extraction Branch. The output from the last TF-GridNet block passes through a transposed convolution layer and an ISTFT module to generate the speech waveform.

All the training is done on a single Nvidia A10 24GB GPU with a batch size of 2. In both training stages, we use the Adam optimizer and decay the learning rate by half when the validation loss does not decrease for more than 50 epochs. Motivated by MOCOv3 [39], which observes that changes in the parameters of shallower layers can lead to instability in the loss, we assign lower learning rates to the earlier layers of the model and higher learning rates to the deeper layers. In particular, the initial learning rate is set to 5e-4 for the whole Siamese Encoder, 1e-3 for the Encoder Fusion Module in the first training stage, and 2e-3 for the whole extraction branch in the second training stage. 500 epochs (200k optimization steps) are used in the first pretraining stage, and 1000 epochs (400k optimization steps) are used in the second stage.

Since TCE is not trained on the LibriSpeech dataset, we fine-tune the TCE model on our simulated data till convergence (for 120k optimization steps) using the Adam optimizer with 5e-4 learning rate. To train and test the TCE model on the proposed extraction task, we modify our dataloader's output to match TCE's application scenario by adding a known speaker's voice in the Negative Enrollment and concatenating the Mixed Audio with this modified Negative Enrollment. The model then performs extraction conditioned on the known speaker's d-vector embedding.

In addition, despite the LookOnceToHear model is trained on audio examples from the LibriSpeech dataset, it uses Scaper toolkit to load and generate audio mixtures. We notice this leads to slightly different audio input to the model if the audio is otherwise loaded by torchaudio.load and mixed by addition. The distribution shift in input audio leads to a noticable difference of around 1 SI-SNRi decrease in performance. For fair comparison, we tune the LookOnceToHear's extraction model on our training data with an initial learning rate set to 5e-4 and decay by half if the validation error does not decrease for more than 50 epochs. The model performance converges after 300 epochs fine-tuning (120k optimization steps) when the lr decreased to around 6e-5.

We load the USEF-TFGridnet model from the open-source project https://github.com/ZBang/USEF-TSE.

Table 7: Model performance with different pooling sizes. The inference time is the average time taken for the model main branch to extract 1-second audio in 50 repeated experiments on an Intel Xeon Silver 4314 @ 2.40GHz CPU.

| Pooling Size | Metric | 2 Speakers in Mixture | | | 3 Speakers in Mixture | | | Inf. Time |
| --- | --- | --- | --- | --- | --- | --- | --- | --- |
| | | 2 Speakers in Enroll | 3 Speakers in Enroll | 4 Speakers in Enroll | 2 Speakers in Enroll | 3 Speakers in Enroll | 4 Speakers in Enroll | |
| 20 | SNRi | 9.57±2.61 | 9.35±2.67 | 9.27±2.79 | 9.96±2.56 | 9.93±2.57 | 9.87±2.68 | 0.365 |
| | SI-SNRi | 8.10±3.90 | 7.82±4.03 | 7.72±4.40 | 7.80±3.99 | 7.60±4.12 | 7.35±4.46 | |
| | STOI | 0.736±0.120 | 0.728±0.125 | 0.713±0.129 | 0.641±0.133 | 0.637±0.131 | 0.627±0.140 | |
| | DNSMOS | 2.02±0.42 | 2.00±0.42 | 1.78±0.40 | **2.00±0.43** | 1.78±0.42 | 1.76±0.41 | |
| 40 | SNRi | **10.14±2.57** | **9.93±2.71** | **9.79±2.86** | **10.42±2.49** | **10.37±2.64** | **10.22±2.79** | 0.36 |
| | SI-SNRi | **8.85±3.67** | **8.54±4.01** | **8.30±4.47** | **8.42±3.62** | **8.29±4.06** | **7.82±4.62** | |
| | STOI | **0.758±0.107** | **0.749±0.121** | **0.742±0.126** | **0.668±0.123** | **0.665±0.130** | **0.648±0.146** | |
| | DNSMOS | **2.14±0.37** | **2.13±0.37** | **2.02±0.38** | 1.93±0.37 | **1.92±0.38** | **1.90±0.38** | |
| 80 | SNRi | 9.93±2.55 | 9.75±2.65 | 9.71±2.74 | 10.18±2.49 | 10.20±2.59 | 10.10±2.72 | **0.33** |
| | SI-SNRi | 8.59±3.77 | 8.28±4.22 | 8.22±4.38 | 8.02±3.82 | 8.00±4.15 | 7.66±4.69 | |
| | STOI | 0.749±0.109 | 0.742±0.122 | 0.739±0.124 | 0.654±0.128 | 0.654±0.133 | 0.642±0.144 | |
| | DNSMOS | 2.09±0.40 | 2.06±0.41 | 1.95±0.44 | 1.86±0.40 | 1.86±0.40 | 1.84±0.40 | |

Table 8: Model performance under different number of Positive Interferers (PI).

| Method | Metric | 2 Speakers in Mixture | | | 3 Speakers in Mixture | | |
| --- | --- | --- | --- | --- | --- | --- | --- |
| | | 1 PI | 2 PI | 3 PI | 1 PI | 2 PI | 3 PI |
| Ours (Monaural) | SNRi | 10.23±2.56 | 10.10±2.58 | 10.05±2.66 | 10.46±2.43 | 10.47±2.54 | 10.49±2.63 |
| | SI-SNRi | 8.98±3.66 | 8.87±3.60 | 8.79±3.76 | 8.53±3.56 | 8.53±3.76 | 8.44±4.09 |
| | STOI | 0.763±0.105 | 0.759±0.106 | 0.758±0.105 | 0.673±0.119 | 0.673±0.122 | 0.667±0.129 |
| | DNSMOS | 2.15±0.37 | 2.14±0.36 | 2.14±0.37 | 1.93±0.37 | 1.93±0.38 | 1.93±0.37 |

In Table 6, we compare our model architecture's parameter size, extraction branch's inference time, and memory usage in inference, with the two prior works on target speaker extraction using noisy enrollments. Our model uses smaller parameter size. In particular, we notice that the Film fusion module substantially increases the parameter size, as shown by the significant increase in model parameter size when we replace our cross-attention fusion module with the Film fusion module.

## D    Embedding Pooling Size

To reduce the model computation time in the Extraction Branch, we perform an average pooling of 40 on the embedding extracted by the encoder head. In this section, we analyze the trade-offs between model performance and inference time under different pooling configurations. Specifically, we compare our model (40 pooling size) with two variants using pooling sizes of 20 and 80, as shown in Table 7.

As the pooling size increases, the length of the extracted target speaker embedding sequence decreases, which reduces the inference time for the extraction branch. Larger pooling size also results in more information loss, which leads to worse extraction performance. To our surprise, a smaller pooling size also results in a worse performance, despite it retains more detailed information of the target speaker in the embedding sequence. We hypothesize that smaller pooling size introduces instability in model training, as the resulting target speaker embedding sequence has larger variation across different timesteps. The unstable model learning could lead to worse final model performance. As a result, we selected 40 pooling sizes as the final model configuration.

## E    Affect of Increasing Number of Positive and Negative Interferers on Model Performance

In Section 4.2 in the main paper, we randomly assign the interferers in the enrollments as Positive or Negative Interferers, and show the model performance as the number of interferers increase. In this section, we separately evaluate how Positive and Negative Interferers will affect model performance. In the first experiment, we include only Positive Interferers in the enrollments and gradually increase their number. In the second experiment, we follow the same procedure but only include Negative

Table 9: Model performance under different number of Negative Interferers (NI).

| Method | Metric | 2 Speakers in Mixture | | | 3 Speakers in Mixture | | |
|---|---|---|---|---|---|---|---|
| | | 1 NI | 2 NI | 3 NI | 1 NI | 2 NI | 3 NI |
| Ours (Monaural) | SNRi | 10.07±2.70 | 9.72±2.85 | 9.33±3.14 | 10.32±2.55 | 10.19±2.73 | 9.93±2.95 |
| | SI-SNRi | 8.69±4.06 | 8.19±4.48 | 7.43±5.39 | 8.23±4.02 | 7.89±4.59 | 7.21±5.36 |
| | STOI | 0.754±0.118 | 0.739±0.129 | 0.717±0.152 | 0.664±0.129 | 0.653±0.142 | 0.630±0.161 |
| | DNSMOS | 2.13±0.38 | 2.10±0.39 | 2.07±0.40 | 1.91±0.38 | 1.89±0.39 | 1.87±0.39 |

Table 10: Model performance under different monaural enrollment lengths. We vary the conditional Positive and Negative Enrollment length between 1 to 10 seconds.

| Condition | Metric | Pos. 1 sec | Pos. 3 sec | Pos. 5 sec | Pos. 10 sec |
|---|---|---|---|---|---|
| Neg. 1 sec | SNRi | 9.64±2.66 | 10.22±2.53 | 10.31±2.51 | 10.43±2.43 |
| | SI-SNRi | 7.07±4.45 | 8.16±3.94 | 8.33±3.78 | 8.54±3.62 |
| | STOI | 0.626±0.142 | 0.662±0.126 | 0.668±0.121 | 0.674±0.117 |
| | DNSMOS | 1.86±0.38 | 1.93±0.38 | 1.94±0.37 | 1.95±0.36 |
| Neg. 3 sec | SNRi | 9.68±2.66 | 10.37±2.64 | 10.38±2.56 | 10.47±2.40 |
| | SI-SNRi | 7.16±4.38 | 8.29±4.06 | 8.39±3.93 | 8.63±3.42 |
| | STOI | 0.629±0.140 | 0.665±0.130 | 0.671±0.122 | 0.677±0.114 |
| | DNSMOS | 1.87±0.39 | 1.92±0.38 | 1.94±0.37 | 1.97±0.36 |
| Neg. 5 sec | SNRi | 9.71±2.70 | 10.26±2.63 | 10.21±2.54 | 10.49±2.41 |
| | SI-SNRi | 7.16±4.64 | 8.17±4.13 | 8.45±3.64 | 8.66±3.46 |
| | STOI | 0.629±0.143 | 0.662±0.128 | 0.671±0.118 | 0.677±0.115 |
| | DNSMOS | 1.87±0.39 | 1.93±0.37 | 1.94±0.37 | 1.96±0.36 |
| Neg. 10 sec | SNRi | 9.74±2.70 | 10.22±2.59 | 10.41±2.43 | 10.50±2.36 |
| | SI-SNRi | 7.28±4.35 | 8.11±4.06 | 8.54±3.47 | 8.71±3.26 |
| | STOI | 0.632±0.136 | 0.660±0.126 | 0.674±0.114 | 0.679±0.110 |
| | DNSMOS | 1.88±0.38 | 1.94±0.37 | 1.95±0.36 | 1.96±0.36 |

Interferers in enrollments. In both cases, we evaluate the model's performance when extracting a target speaker from Audio Mixtures containing 2–3 speakers.

As shown in Table 8 and Table 9, our model shows only 0.2 dB SI-SNRi decrease in performance as the number of Positive Interferer increases from 1 to 3. In comparison, when the number of Negative Interferers increase, the model shows more significant performance decrease of over 1 dB SI-SNRi. This suggests that, in comparison to removing interfering speakers' characteristic given in the Negative Enrollment, our model is more capable of identifying the target speaker's characteristic, which is a common voice characteristic that exists across different segments in the Positive Enrollment.

## F  Model Performance under Different Lengths of Positive and Negative Enrollment

We train our model solely on samples with 3 seconds of Positive and Negative Enrollments, and tested the model performance solely on these scenarios. In this section, we test the model performance on different enrollment lengths between 1 to 10 seconds. We keep the ratio of the Positive Interferer in the Positive Enrollment, and the ratio of the Negative Interferer in the Negative Enrollment unchanged. This means the Positive Enrollment will still span $1/3$ to $2/3$ of the Positive Enrollment, and Negative Interferer span the $1/3$ to full length of the Negative Enrollment. As shown in Table 10, the model performance improves in general as both the length of the Positive Enrollment and the Negative Enrollments increase. We also notice that increasing the Negative Enrollment length when less than 3 seconds of the Positive Enrollment is provided could adversely affect the model performance. This might be because the model has too little information of the target speaker

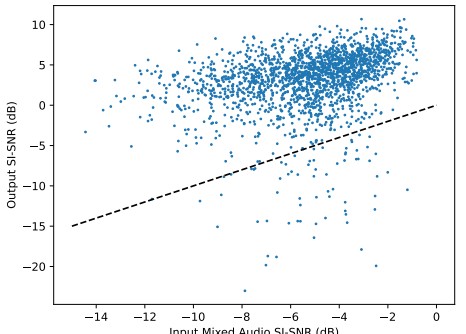

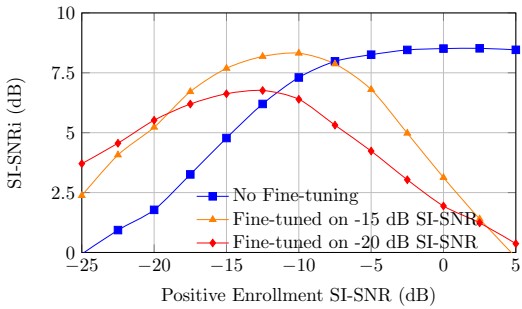

Figure 6: SI-SNR of the extracted audio with respect to the Mixed Audio. The dashed black line represents the zero-improvement line.

Figure 7: SI-SNRi of the extracted audio with respect to the Positive Enrollment SI-SNR.

Table 11: Model performance under different number of Hybrid Interferers (HI)

| Method | Metric | 2 Speakers in Mixture | | | 3 Speakers in Mixture | | |
|---|---|---|---|---|---|---|---|
| | | 1 HI | 2 HI | 3 HI | 1 HI | 2 HI | 3 HI |
| | SNRi | 10.23±2.55 | 10.06±2.59 | 10.05±2.63 | 10.47±2.43 | 10.49±2.52 | 10.53±2.60 |
| Ours | SI-SNRi | 8.99±3.63 | 8.82±3.61 | 8.83±3.58 | 8.54±3.56 | 8.57±3.70 | 8.53±3.92 |
| (Monaural) | STOI | 0.763±0.105 | 0.757±0.108 | 0.759±0.103 | 0.673±0.119 | 0.674±0.119 | 0.669±0.126 |
| | DNSMOS | 2.15±0.37 | 2.14±0.36 | 2.14±0.37 | 1.93±0.37 | 1.93±0.38 | 1.93±0.37 |

form the Positive Enrollment, while the Negative Enrollment might guides the model to remove excessive speaker characteristics from the Positive Enrollment, resulting in the decrease in model performance. This suggests that users can enhance extraction quality by providing longer Positive and Negative enrollments if the results are unsatisfactory, and increasing the Positive Enrollment length is preferable.

## G    Model Performance with respect to the Input Audio Quality

**Model Performance with respect to the Audio Mixture Quality**    In this section, we show our model performance relative to the quality of input Audio Mixture. As shown in Figure 6, we plot the extracted audio's SI-SDR values for 5000 randomly generated test samples as a function of their input SI-SDR. 96.6% of the audio mixtures' SI-SNR were improved after the extraction.

**Model Performance with respect to the Positive Enrollment Quality**    Figure 7 presents our model's extraction performance as a function of enrollment audio quality. To evaluate this, we scale the clean enrollment audio of the target speaker (Positive Enrollment) to specific SI-SNR values, while leaving the Negative Enrollment and the Mixed Audio unchanged. We then measure the average of 5000 test samples' extracted audios' SI-SNRi at each Positive Enrollment SI-SNR value.

The model achieves optimal performance when the Positive Enrollment SI-SNR is between -5 dB to 5 dB, and the performance deteriorates significantly when the SI-SNR of the Positive Enrollment falls below -10 dB. To address this limitation, we fine-tune our model by scaling the training data's Positive Enrollment to -20 and -15 SI-SNR dB. As shown by the red line and the orange line in Figure 7, this fine-tuning shifts the model's peak performance to lower SI-SNR regions, which means the model achieves optimum extraction performance under worse Positive Enrollment quality. Further research is necessary to develop a model with strong performance across the full SI-SNR range of the Positive Enrollment.

## H    Model Performance under presence of Hybrid Interferers

As defined in Section 3.2 in the main paper, when an interfering speaker speaks in a subset of segments in the Positive Enrollment, and also exist in the Negative Enrollment, it is a Hybrid Interferer. Model

Table 12: Model performance under different number of Neglect-Required Interferers (NRI).

| Method | Metric | 2 Speakers in Mixture | | | 3 Speakers in Mixture | | |
|---|---|---|---|---|---|---|---|
| | | 1 NRI | 2 NRI | 3 NRI | 1 NRI | 2 NRI | 3 NRI |
| Ours (Monaural) | SNRi | 10.29±2.52 | 10.20±2.54 | 10.25±2.55 | 10.45±2.44 | 10.51±2.49 | 10.57±2.54 |
| | SI-SNRi | 9.07±3.55 | 8.99±3.56 | 9.08±3.43 | 8.51±3.60 | 8.63±3.56 | 8.63±3.77 |
| | STOI | 0.764±0.103 | 0.761±0.105 | 0.764±0.098 | 0.672±0.120 | 0.675±0.117 | 0.671±0.125 |
| | DNSMOS | 2.15±0.36 | 2.14±0.35 | 2.15±0.36 | 1.93±0.37 | 1.93±0.38 | 1.92±0.37 |

Table 13: Comparison between our method and the TD-SpeakerBeam baselines' performance when extracting conditioned on clean or noisy enrollments.

| Method | Metric | 2 Speakers in Mixture | | 3 Speakers in Mixture | | 4 Speakers in Mixture | |
|---|---|---|---|---|---|---|---|
| | | Clean Enroll | Noisy Enroll | Clean Enroll | Noisy Enroll | Clean Enroll | Noisy Enroll |
| Ours (Monaural) | SNRi | 10.19±2.59 | **9.93±2.71** | **10.48±2.46** | **10.37±2.64** | **10.48±2.35** | **10.50±2.52** |
| | SI-SNRi | 9.12±3.50 | **8.54±4.01** | **8.50±3.63** | **8.29±4.06** | **7.79±3.70** | **7.64±4.19** |
| | STOI | 0.764±0.102 | **0.749±0.121** | **0.670±0.122** | **0.665±0.130** | **0.592±0.131** | **0.584±0.141** |
| | DNSMOS | 2.15±0.37 | **2.13±0.37** | 1.93±0.37 | **1.92±0.38** | 1.76±0.37 | **1.65±0.40** |
| TD-SpeakerBeam | SNRi | **12.87±3.63** | 4.84±2.83 | 10.01±3.99 | 5.58±2.48 | 9.08±2.69 | 6.63±2.60 |
| | SI-SNRi | **11.85±6.36** | 1.75±11.89 | 7.63±6.06 | -0.33±8.70 | 5.49±5.34 | -1.26±7.17 |
| | STOI | **0.834±0.145** | 0.595±0.270 | 0.654±0.182 | 0.451±0.212 | 0.534±0.177 | 0.372±0.177 |
| | DNSMOS | **2.81±0.32** | 1.73±0.59 | **2.63±0.32** | 1.51±0.40 | **2.45±0.35** | 1.46±0.48 |

have two ways to distinguish these interfering speakers from the target speaker: either through noticing that these speakers were silent in some of the segments in the Positive Enrollment, when the target speaker is supposed to talk continuously; or noticing that these speakers exist in the Negative Enrollment, when the target speaker is supposed to remain silent. In this section, we explicitly show the model performance when such interfering speakers exist.

As shown in Table 11, we focus on the scenario containing only Hybrid Interferers, and gradually increase their number from 1 to 3. In comparison to the increase in the Positive and Negative Interferer number, our model shows higher robustness to the increase in the number of Hybrid Interferers. This indicates that our model generalizes well in removing interference from Hybrid Interferers by leveraging the fact that such speakers can be effectively suppressed when treated as either Positive or Negative Interferers.

## I  Model Performance under presence of Neglect-Required Interferers

When recording the Negative Enrollments, additional interfering speakers not captured in the Positive Enrollments might be present. Since these interferers' speech does not overlap with the target speaker, their voice should be neglected (thus named neglect-Required Interferers). In this section, we investigate their affect on our model performance in this section.

As shown in Table 12, similar to the result observed when the number of Hybrid Interferers increase, our model does not show statistically significant performance differences when the number of Neglect-Required Interferers increases from one to three. This demonstrates our model effectively generalizes to prevent the Neglect-Required Interferers from adversely affecting the quality of the extracted target speaker speech.

## J  Model Performance when Clean Target Speaker's Enrollment is available

If applied without modification, our model fails when the enrollment consists solely of the clean target speaker's voice. This is due to the model's input normalization, where an all-zero Negative Enrollment leads to a NaN output. To address this, we randomly select a fixed WHAM! noise sample as the Negative Enrollment, and add it with the Positive Enrollment to construct a pseudo-noisy Positive Enrollment. Using such modification, we construct pseudo-Positive and Negative Enrollment from different target speakers' clean audio enrollment, and perform extraction.

As shown in Table 13, in comparison to extracting target speaker's characteristic from noisy Enrollments, our model achieves higher performance when extracting conditioned on the pseudo-Positive

Table 14: Model performance when extracting conditioned on noisy single-speaker enrollments.

| Method | Metric | 2 Speakers in Mixture | 3 Speakers in Mixture |
|--------|--------|------------------------|------------------------|
| Ours (Monaural) | SNRi | **10.19±2.59** | **10.48±2.46** |
| | SI-SNRi | **9.12±3.50** | **8.50±3.63** |
| | STOI | **0.76±0.10** | **0.67±0.12** |
| | DNSMOS | **2.15±0.37** | **1.93±0.37** |
| USEF-TFGridnet | SNRi | 3.54±3.45 | 4.29±2.46 |
| | SI-SNRi | 0.50±5.58 | 0.56±2.85 |
| | STOI | 0.47±0.17 | 0.38±0.11 |
| | DNSMOS | 1.35±0.44 | 1.29±0.37 |

Table 15: Speaker configuration for evaluating the Target Speaker Confusion Problem.

| | Target Speaker | Interfering Speaker |
|--|----------------|---------------------|
| Audio Mixture | A | B |
| $P_{pred}$ | A | B |
| $P_{tgt}$ | A | B |
| $P_{interfer}$ | B | A |

and Negative Enrollment. In comparison to the TD-SpeakerBeam [5], which is trained to extract using clean audio enrollment, our model achieves higher SNRi, SI-SNRi, and STOI when extracting from the mixture of 3 or more mixtures. These results verifies our model's applicability to extraction using clean audio enrollments. The reason why our model achieves higher performance under larger number of interfering speaker might be the difference in training data generation. TD-SpeakerBeam is trained on predefined mixtures of different speakers' speech, while our training samples are generated by dynamic mixing, which creates more diverse mixture of different speakers' voices. This training data generation strategy acts as a form of data augmentation, helping the model generalize better to larger number of interfering speaker. However, there is still noticeable performance gap between our model and the TD-SpeakerBeam when extracting from the mixture of two speakers using clean enrollments. Further work is required to reduce the performance gap between models using noisy enrollment and using clean enrollment.

We want to emphasize that our model tackles the more challenging task of target speech extraction (TSE) with noisy enrollment, a realistic scenario where prior works have shown significantly degraded performance. Although our model has worse performance under clean audio enrollment, our method still correctly identifies the target speaker users want, shown by the strongly positive SI-SNRi values. Indeed, the performance gap between our method and other models under clean enrollment could bring novel insights to improve our model, which we will explore in the further works.

In addition, we evaluate our model and the baseline models' performance when using noisy single speaker enrollments. The noise in the enrollment is from the WHAM! dataset and set at 0 SNR level. As shown in Table 14, TSE models trained with clean enrollments show significant performance decrease under noisy single speaker enrollments. In comparison, our model consistently outperforms the baselines across all metrics except DNSMOS, demonstrating greater robustness to enrollment noise.

## K  Model Performance for Target speaker Confusion Problem

Prior works have shown that the auxiliary speaker encoder may sometimes generate ambiguous speaker embeddings, leading to the extraction model extracting interferer's voice as the output. This is known as the target confusion problem [40]. To further investigate whether our model suffers from the target speaker confusion problem, we follow the experimental setup proposed by Zhao et al. [40]. Specifically, we construct 5000 test samples where both the audio mixture and the enrollment contain two speakers, and share the same interfering speaker. Let $P_{pred}$ be the enrollment pair in the sample, where speaker A is the target speaker and speaker B is the interfering speaker. After performing extraction, we identify the enrollment pairs that result in the extracted audio being more similar (in terms of SNR) to the interfering speaker B's speech than to the target speaker A's. We refer to these samples as target confusion samples. For each target confusion sample, we construct two additional enrollment pairs: $P_{tgt}$ and $P_{interfer}$. In $P_{tgt}$, same as the target confusion sample, speaker A is the target and speaker B is the interferer. In $P_{interfer}$, speaker B is the target and speaker A is the interferer. Table 15 below summarizes the role of speaker A and speaker B in the audio mixture and each enrollment pair.

Table 16: Performance of our model and the TCE model before and after fine-tuning, using monaural samples generated from the WSJ0 dataset.

| Method | Metric | 2 Speakers in Mixture | | | 3 Speakers in Mixture | | |
|---|---|---|---|---|---|---|---|
| | | 1 NI | 2 NI | 3 NI | 1 NI | 2 NI | 3 NI |
| Ours (Monaural) | SNRi | **8.33±2.97** | **8.14±3.03** | **7.93±3.04** | 8.69±2.64 | 8.50±2.73 | 8.24±2.81 |
| | SI-SNRi | **5.66±6.04** | **5.33±6.25** | **4.79±6.52** | 5.04±6.03 | 4.51±6.34 | 3.84±6.63 |
| | STOI | **0.672±0.164** | **0.664±0.171** | **0.647±0.181** | 0.591±0.172 | **0.577±0.177** | **0.554±0.187** |
| | DNSMOS | **1.93±0.39** | **1.91±0.40** | **1.89±0.41** | 1.74±0.37 | **1.72±0.37** | **1.69±0.37** |
| Ours (Fine-tuned) | SNRi | 8.19±3.00 | 7.97±3.07 | 7.75±3.06 | **8.96±2.70** | **8.80±2.73** | **8.58±2.88** |
| | SI-SNRi | 5.33±6.17 | 4.93±6.47 | 4.45±6.50 | **5.27±6.17** | **4.84±6.40** | **4.20±6.85** |
| | STOI | 0.647±0.177 | 0.640±0.184 | 0.621±0.190 | 0.580±0.181 | 0.567±0.188 | 0.544±0.198 |
| | DNSMOS | 1.87±0.46 | 1.84±0.46 | 1.81±0.47 | 1.70±0.42 | 1.68±0.42 | 1.65±0.41 |
| TCE | SNRi | 6.65±2.79 | 6.20±2.81 | 5.79±2.95 | 6.64±2.79 | 6.43±2.35 | 6.11±2.39 |
| | SI-SNRi | 4.00±4.93 | 3.13±5.28 | 2.21±5.87 | 3.99±4.93 | 2.19±4.98 | 1.46±5.20 |
| | STOI | 0.593±0.149 | 0.567±0.158 | 0.546±0.169 | **0.593±0.149** | 0.471±0.156 | 0.453±0.159 |
| | DNSMOS | 1.74±0.38 | 1.73±0.37 | 1.75±0.37 | **1.74±0.38** | 1.60±0.35 | 1.63±0.35 |
| TCE (Fine-tuned) | SNRi | 6.59±2.62 | 6.20±2.69 | 6.03±2.81 | 6.58±2.62 | 6.54±2.33 | 6.34±2.41 |
| | SI-SNRi | 3.99±4.41 | 3.21±4.67 | 2.67±5.25 | 3.97±4.41 | 2.55±4.43 | 2.10±4.62 |
| | STOI | 0.576±0.146 | 0.548±0.154 | 0.533±0.163 | 0.576±0.146 | 0.462±0.153 | 0.446±0.154 |
| | DNSMOS | 1.59±0.40 | 1.56±0.39 | 1.57±0.38 | 1.59±0.40 | 1.46±0.34 | 1.46±0.33 |

To verify if the model mistakenly encode the interfering speaker as the target speaker and the interfering speaker, we flatten and normalize the embeddings extracted by our model from each enrollment pair, obtaining $E(P_{pred})$, $E(P_{tgt})$, $E(P_{interfer})$, and compute the cosine similarity between $E(P_{pred})$ and $E(P_{tgt})$, as well as between $E(P_{pred})$ and $E(P_{interfer})$.

Out of the 5000 tested samples, 74 of them are the target confusion samples. 34 samples out of the 74 target confusion samples (45.9%) have enrollment embedding closer to the interfering speaker than to the target speaker (i.e. 34 of the samples have $cos(E(P_{pred}), E(P_{tgt})) < cos(E(P_{pred}), E(P_{interfer}))$). These findings are consistent with those reported in the target speaker confusion study [40], where 45.1% of target confusion samples showed the extracted embedding being closer to the interferer. This means that significant percentage of target confusion samples have encoder embeddings being more close to the interfering speaker. However, it is important to emphasis that only 74 out of the 5000 samples (1.48%) showed evidence of target speaker confusion in our evaluation, suggesting that this is a rare occurrence and does not pose a significant issue for our model's overall performance.

## L    Model performance on WSJ0 Dataset

In the main paper, we focus our experiments on the samples constructed from the LibriSpeech dataset. In this section, we show our model's applicability to other datasets. We select the WSJ0 dataset as the source of speaker speech.

As shown in Table 16, our model correctly encode and extract the target speaker's voice, as shown by the positive SNRi and SI-SNRi metric, and outperforms the TCE baseline in most scenarios. However, after fine-tuning on samples containing two interfering speakers in both the Audio Mixture and enrollments, our model only improves performance when extracting from three-speaker mixtures, while performance degrades in two-speaker cases. Further exploration is required to train a model with strong generalizability across multiple datasets and scenarios.

## M    Audio Visualization and Failure Cases

In this section, we show four successful cases and four failing cases of our model. For each case, we visualize the waveform and STFT spectrogram of the Mixed Audio, the extraction ground truth, and the model prediction of a one-second audio segment.

As shown in Figure 8, our model correctly predicts silent sound in the extracted audio when the target speaker is not speaking and extracts the target speaker's voice despite the frequency range largely overlaps with the interfering speakers.

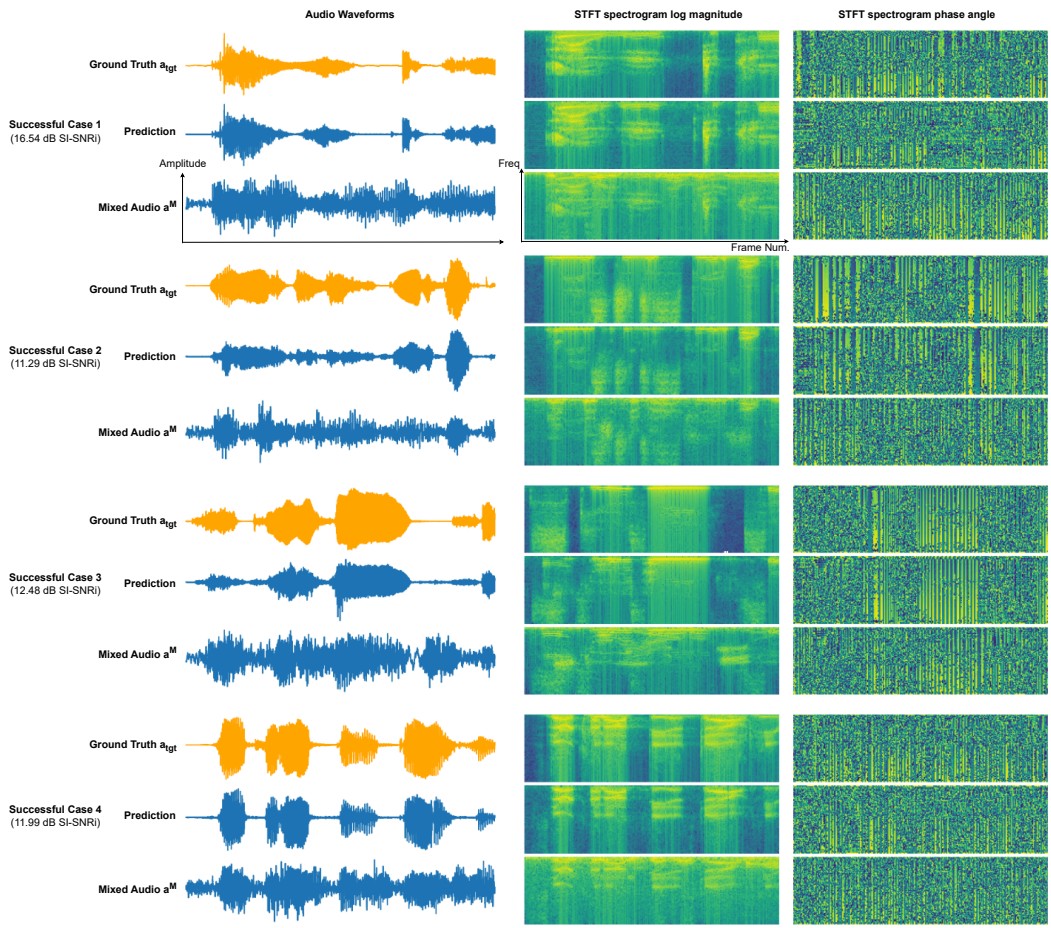

Figure 8: Successful cases of our model. We selected four samples where the model SI-SNRi performance is above 11 dB. We report the extracted audio SI-SNRi under the case number.

In several cases, our model has less optimum performance, with less than 7 dB SI-SNRi performance. We show four of these cases in Figure 9. In these cases, the model fails to extract the full target speaker's voice in some timesteps (Failure Cases 2), includes interfering the speaker's voice in its extraction (Failure Case 1, 3, 4), and introduces artifacts that obscure the intelligibility of the target speech. These non-exhaustive failure cases serve as a reference for future improvements.

## N  Societal Impact

This paper presents work whose goal is to advance the field of Machine Learning and Audio Processing. This technology can potentially improve communication aids, hearing devices, and audio editing tools, making it easier for users to focus on desired voices in complex auditory scenes. We acknowledge that such technology could raise privacy concerns if misused to extract a speaker's voice without consent. However, our method requires prior access to the target speaker's speech activity labels, which presupposes that the target speaker is already engaged in a face-to-face conversation or otherwise clearly audible to the user. As a result, our approach serves to enhance human auditory perception rather than introduce new capabilities that could pose societal or ethical risks. While our work may have broader societal implications, we do not identify any that require specific emphasis.

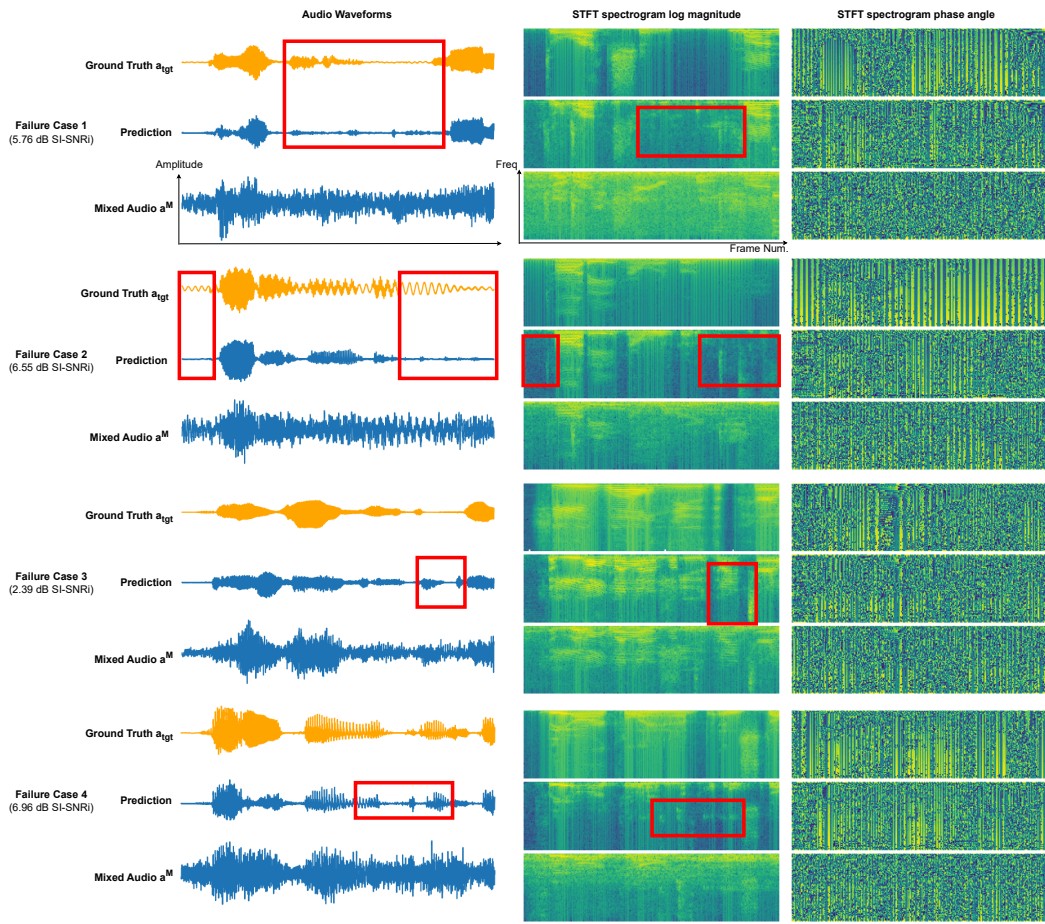

Figure 9: Failure cases of our model. We selected four samples where the model SI-SNRi performance is below 8 dB. We highlight the timesteps where the model has poor performance with red boxes.

