# OpenReview forum: "Target Speaker Extraction through Comparing Noisy Positive and Negative Audio Enrollments"
_NeurIPS.cc/2025/Conference — NeurIPS 2025 poster_

### Official Review · Reviewer_5AuH · 2025-06-24

**Clarity:** 3
**Significance:** 3
**Originality:** 4
**Rating:** 5
**Confidence:** 4

**Summary:**

This paper addresses the TSE problem in a challenging setting: the target user cannot provide their enrollment audio in a clean acoustic environment, but have to provide samples in the noisy condition with background noise and interference speakers. It is a very interesting topic in speech that's rarely discussed, but is of actually good use cases in reality. The contribution is mainly about training an Encoder Fusion Module in the first stage, then training an Extraction Fusion Module in a second stage. The second stage is pretty standard. The first stage is very interesting.

**Questions:**

What is the mechanism to remove Positive Interferer (PI) in the Encoder Fusion Module? This part is very unclear to me. Both the model architecture or the loss function of stage 1 do not seem to have a built-in mechanism to remove positive interferer. I'd like to see more clarification or explanation on this.

Line 173: I assume T_pos is number of frames of positive enrollment audio? Worth clarifying.

**Ethical Concerns:**

["NO or VERY MINOR ethics concerns only"]

**Final Justification:**

The authors have addressed all of my questions. My final rating is 5.

**Limitations:**

The real limitation of this work is that it is not evaluated with any realistic data. This should be clearly stated.

**Paper Formatting Concerns:**

No concern.

**Quality:**

4

**Strengths And Weaknesses:**

Strengths:

Paper is very well written. Easy to read in general except for the Encoder Fusion Module section. Figure 1 and Figure 2 are super helpful.

Ablation studies are very well designed, and answered important questions, i.e. Effectiveness of the two-stage model training, and Cross-Attention based Fusion over Film Fusion Method. The experiments on inaccurate Positive and Negative Labeling are very helpful.

The Encoder Fusion Module work (stage 1) is novel. I never saw similar work.

The assumption about the start/end time being unique is easy to satisfy in reality.


Weaknesses:

Line 175: For the entire "The Encoder Fusion Module" section, this is the most important part, but it is a bit difficult to follow. I would suggest using equations wherever possible. Also, consider adding a new diagram explaining how this module works.

"Model performance under inaccurate Positive and Negative Labeling" - it is really great to have this section. But I think it can be improved. First, 0.30 ± 0.19 seconds do not seem to be super inaccurate. What if the speaker boundary errors are above 1 second? Also, I am not sure I understand Figure 6. What is x-axis? Each column is one sample? Why the red square can be sometimes high (sample 7) and sometimes low (sample 10)? Please consider remaking this Figure or replace with a table.

All the experiments are based on synthetic datasets. Although it is common, it's not ideal, especially this paper aims to addess a special use case in real world applications.

---

> ### Author Rebuttal · Authors · 2025-07-29
>
> **Q1**: Line 175: For the entire "The Encoder Fusion Module" section, this is the most important part, but it is a bit difficult to follow. I would suggest using equations wherever possible. Also, consider adding a new diagram explaining how this module works. Line 173: I assume T_pos is number of frames of positive enrollment audio? Worth clarifying.
>
> **A1**: Thank you for your suggestions. We provide further explanation on the Fusion module in the response to reviewer iQ1f Q5, and will include the pseudo-code of the module in the revised version. T_pos indeed represents the number of STFT frames in the Positive Enrollment.
>
>
> **Q2**: "Model performance under inaccurate Positive and Negative Labeling" - it is really great to have this section. But I think it can be improved. First, 0.30 ± 0.19 seconds do not seem to be super inaccurate. What if the speaker boundary errors are above 1 second? Also, I am not sure I understand Figure 6. What is x-axis? Each column is one sample? Why the red square can be sometimes high (sample 7) and sometimes low (sample 10)? Please consider remaking this Figure or replace with a table.
>
> **A2**: Thank you for your suggestions. In the user study on inaccurate positive and negative labeling, among the 100 user-provided labels across 10 samples, 18 labels for audio samples [1, 3, 4, 6, 10] deviate more than one second from the ground truth boundaries. We report the average model SI-SNRi on these inaccurate labels in the table below. Sample [1,3,4,6] all show less than 1 dB SI-SNRi performance degradation compared to the model performance using ground truth boundaries, despite a temporal labeling inaccuracy of 1.36 $\pm$ 0.44 seconds. For sample 10, only one user label deviates by more than one second, reaching 4.1 seconds. This results in 5.16 dB SI-SNRi. Overall, these results demonstrate the robustness of our model to inaccuracies in user-provided boundary annotations.
>
> | sample ID  | 1 | 3 | 4 | 6 | 10 |
> |----------------|--------|-------|-------|--------|--------|
> | Ground Truth Labeling    | 10.98 | 8.38 | 11.71 |12.31 | 11.81 |
> | User Labeling (> 1s deviation) | 10.89 | 8.35 | 11.62 | 11.90 | 5.16 |
>
> Regarding Figure 6, you are correct that the x-axis represents the sample ID, and each column corresponds to the model performance on an individual sample. The red markers indicate outliers, which appear above or below the box plot when a model’s performance on one of the 10 user labelings for that sample exceeds the upper quartile or falls below the lower quartile, respectively. To improve clarity, we will revise Figure 6 as a table reporting the mean and standard deviation of model performance for each sample.
>
> **Q3**: All the experiments are based on synthetic datasets. Although it is common, it's not ideal, especially this paper aims to addess a special use case in real world applications.
>
> **A3**: Thank you for mentioning this. Indeed, to effectively demonstrate our model's performance under strong overlap between the target and interfering speakers, all audio mixtures used in our original submission were synthetically generated or manually added up from real-world audio recordings. We additionally compare our model and the baseline methods on naturally recorded real-world audio mixtures. Please refer to the response to reviewer 1MM1 Q1 for the result.
>
> **Q4**: What is the mechanism to remove Positive Interferer (PI) in the Encoder Fusion Module? This part is very unclear to me. Both the model architecture or the loss function of stage 1 do not seem to have a built-in mechanism to remove positive interferer. I'd like to see more clarification or explanation on this.
>
> **A4**: Sorry for the confusion. We based the model Encoder Fusion Module on self-attention layers. The self-attention operation allows model to perform pairwise comparison between the embeddings of different frames in the Positive Enrollment. As a result, this architecture design allows the model to identify the speakers who do not talk in some of the Positive Enrollment frames and exclude them in the extracted embedding. To guide the model exclude PI's characteristic in its embedding, we simulated the Positive Enrollments so that the number of PI varies between 0 to 2 in the stage 1 training. Distilling the encoder to only encode the target speaker's voice in the Positive Enrollment forces the model to exclude the PI's voice characteristics in its extraction. Please see the response to reviewer iQ1f Q5 for more details.

---

> > ### Comment · Reviewer_5AuH · 2025-08-04
> >
> > The pseudo code for the Encoder Fusion Module is very helpful, but also raises more questions...
> >
> > "Elementwise add segmentation embedding via broadcasting" - how are these segmentation embeddings initialized? Random? Do you have any loss function to penalize these segmentation embeddings to become too similar?
> >
> > "Apply two Full-band Self-attention calculation" - what do you mean by full-band? I don't think this is a widely accepted term in the ML community. Do you mean the dimensions of the key/value/query are the same as the input? Or there is no projection on the value? Please clarify. Also, why does it have to be full-band?

---

> > > ### Author Response · Authors · 2025-08-07
> > >
> > > Dear Reviewer,
> > >
> > > Thank you again for your thoughtful comments and questions. We have provided further explanations to your questions in our previous response. In summary, the training methods and model architectures of the components in Encoder Fusion Module follows the design principles from prior works, while our integration and application of these elements demonstrate their effectiveness in extracting target speaker characteristics in a novel way.
> > >
> > > We sincerely appreciate your time and feedback, and we are happy to provide further clarification if there are any remaining concerns.

---

> ### Author Response · Authors · 2025-08-05
>
> Sorry for the confusion, we provide additional clarification on the Encoder Fusion module:
>
> **Q1**: How are the segmentation embeddings initialized? Do you have any loss function to penalize these segmentation embeddings to become too similar?
>
> **A1**: We realize that the term "segmentation embedding" used in our paper refers to what is commonly known as "segment embedding" in Transformer-based models such as BERT. We apologize for the terminological inconsistency and will revise the terminology accordingly in the final version to avoid confusion.
>
> The segment embeddings in our model are randomly initialized and jointly trained with other learnable parameters in the Encoding Branch. There is no additional loss term penalizing the similarity between the segment embeddings of the Positive and Negative Enrollments. This design choice follows the standard practice in BERT-based models[1, 2, 3], which also train randomly initialized segment embeddings without explicit constraints.
>
> **Q2**: What do you mean by full-band, and why does the Encoder Fusion Module have to be full-band?
>
> **A2**: As noted in line 161 of the original paper, the term "Full-band Self-attention" originates from the prior work TFGridnet[4]. Since this module was directly adopted from the prior work, we simplified the explanation regarding the shapes of the positive and negative enrollment embeddings. We provide those details here for clarity.
>
> For simplicity, we referred to the Positive and Negative Enrollment embeddings as having shape $[T_{\text{pos}}, D]$ and $[T_{\text{neg}}, D]$. Their actual shapes are $[T_{\text{pos}}, F, E]$ and $[T_{\text{neg}}, F, E]$, respectively, where $T_{\text{pos}}$ and $T_{\text{neg}}$ denote the number of time frames, $F$ is the number of frequency bins, and $E$ is the embedding dimension for each time-frequency unit. After the elementwise-addition with the $[F, E]$ shaped segment embeddings and concatenation, the input to the Full-band Self-attention module has shape $[T_{\text{pos}} + T_{\text{neg}}, F, E]$.
>
> The Full-band Self-attention module flattens the embedding and frequency dimension, resulting in a tensor of shape $[T_{\text{neg}} + T_{\text{pos}}, F * E]$ before applying self-attention. The term *Full-band* reflects that the $E*F$ dimensional embedding vector of each frame encodes **all the frequency bands**, allowing the attention mechanism to operate over full spectral information.
>
> We adopt this module in the Encoder Fusion Module because the task of fusing positive and negative enrollment embeddings requires discriminating between different speakers' characteristics, which are distributed across a wide frequency range. A module that integrates information from the entire frequency range is therefore better suited to capture speaker-specific cues that might lie in different frequency bands.
>
> We apologize again for the unclear explanation and the inconsistent terminology, and hope the above clarification resolves your questions. We are happy to provide further clarification if there are any remaining concerns.
>
> **References:**
>
> [1] Z. Lan, M. Chen, S. Goodman, K. Gimpel, P. Sharma, and R. Soricut, “ALBERT: A Lite BERT for self‑supervised learning of language representations,” in Proc. International Conference on Learning Representations (ICLR), 2020.
>
> [2] Weijie Su, Xizhou Zhu, Yue Cao, Bin Li, Lewei Lu, Furu Wei, and Jifeng Dai, “VL‑BERT: Pre‑training of Generic Visual‑Linguistic Representations,” in Proc. International Conference on Learning Representations (ICLR), 2020.
>
> [3] J. Devlin, M.‑W. Chang, K. Lee, and K. Toutanova, “BERT: Pre‑training of deep bidirectional transformers for language understanding,” in Proc. NAACL‑HLT, 2019, pp. 4171–4186.
>
> [4] Z.-Q. Wang, S. Cornell, S. Choi, Y. Lee, B.-Y. Kim, and S. Watanabe, “TF‑GridNet: Integrating Full‑ and Sub‑Band Modeling for Speech Separation,” IEEE/ACM Trans. Audio, Speech & Lang. Process., 2023.

---

### Official Review · Reviewer_1MM1 · 2025-06-29

**Clarity:** 4
**Significance:** 3
**Originality:** 3
**Rating:** 5
**Confidence:** 4

**Summary:**

This paper studies the target speaker extraction (TSE) problem where the task is to extract the target speaker's voice from an audio mixture provided that some enrollment audio segments are given as the conditional input to the TSE model. Different from prior works that typically assume availability of the clean speech of the target speaker for the enrollment audio, this paper focuses on the more realistic scenario where the enrollment audio segments can contain both the target speaker's voice and interfering speakers' speech as well as background noise. The main idea is to utilized positive and negative noisy enrollments by exploiting the temporal misalignment of speech signals. To this end, a two-branch encoder design and an associated two-stage training strategy are introduced to effectively learn the extraction functionality. Experimental results on synthetic audio mixture data are provided to demonstrate the effectiveness of the proposed method over existing TSE models that have not sufficiently leveraged noisy enrollments as the conditioning input for realistic use cases.

**Questions:**

1. The loss function for training the extraction branch, i.e., eq. (3) seems relatively simple -- it is just the SNR between the ground truth signal and the output of the TSE model. I wonder if this would lead to sub-optimal performance and what motivated you to use this loss? Also, have you experimented with other more sophisticated loss functions, e.g., time-frequency domain MSE losses or some other perceptually motivated losses?

2. Will you make the code public if the paper is accepted?

**Ethical Concerns:**

["NO or VERY MINOR ethics concerns only"]

**Final Justification:**

During the rebuttal period, the authors provided additional evaluation results on naturally recorded real-world audio mixtures, which addressed my earlier concerns about the method’s performance in realistic scenarios. The accompanying subjective evaluation further supports the effectiveness of their approach. I am therefore happy to raise my score to Accept.

**Limitations:**

yes

**Paper Formatting Concerns:**

I have found no formatting issues in this paper.

**Quality:**

3

**Strengths And Weaknesses:**

**Strengths**

1. The problem that the paper aims to address is interesting and important, i.e., TSE with noisy enrollments, which is an essential step towards more realistic scenarios and applications of target speaker extraction.

2. The main idea of exploiting the temporal misalignment of voices from different speakers seems novel and makes sense, which also leads to effective TSE with noisy enrollments.

3. Extensive ablation studies on one-stage vs. two-stage training schemes, different enrollment segment lengths, accuracy of the labeling, etc. offer further insights into understanding the behavior of the proposed approach.

4. The paper is well-written and easy to follow, with code and audio samples provided to facilitate reproducibility and qualitative comparison.

**Weaknesses**

1. The experimental results are mostly relying on artificially mixed audio data. Given that this paper aims to address the limitation of prior works for real-world TSE applications, it is expected to see more evaluation results of the method on naturally recorded mixture audio, to further confirm its effectiveness. Although there are some relevant results presented in Appendix K, it seems that the mixture signals in the experiment are still manually added up. In addition, there is no comparison with other existing approaches.

---

> ### Author Rebuttal · Authors · 2025-07-29
>
> **Q1**: The experimental results are mostly relying on artificially mixed audio data. Given that this paper aims to address the limitation of prior works for real-world TSE applications, it is expected to see more evaluation results of the method on naturally recorded mixture audio, to further confirm its effectiveness. Although there are some relevant results presented in Appendix K, it seems that the mixture signals in the experiment are still manually added up. In addition, there is no comparison with other existing approaches.
>
> **A1**: Thank you for raising this question. To effectively demonstrate our model's performance under strong overlap between the target and interfering speakers, all audio mixtures used in our original submission were synthetically generated or manually added up from real-world audio recordings.
>
> We additionally compare our model and the baseline methods on naturally recorded real-world audio mixtures. We evaluate the Mean Opinion Score (MOS) of different models using five naturally recorded audio mixtures sourced from Freesound.org and the VoxConverse dataset. These sounds are noisy real-world speech recordings captured in pubs, metro stations, urban areas, and city council meetings. For the evaluation, we manually labelled the Positive and Negative Enrollment and extracted target speech from Audio Mixture clips taken from the same recording. The duration of the enrollment and Audio Mixture clips range from 3 to 8 seconds.
>
> 10 participants are asked to rate the original (unprocessed) Audio Mixture, and the audio extracted by three models. Given a textual description of the target speaker (e.g., the female speaker talking over the crowd), they rated the clarity of the target speech relative to background noise on a 1–4 scale. As shown in the table below, our model achieved a MOS of 3.35, outperforming all baseline methods. These results demonstrate the superiority and practical applicability of our model in real-world target speech extraction scenarios.
>
> | Model | Average User Mark |
> |-|-|
> | Ours (Monaural) | 3.35 |
> | Ours (FiLM Fusion) | 2.60 |
> | USEF-TFGridnet | 1.45 |
> | Unprocessed | 2.10 |
>
> **Q2**: The loss function for training the extraction branch, i.e., eq. (3) seems relatively simple -- it is just the SNR between the ground truth signal and the output of the TSE model. I wonder if this would lead to sub-optimal performance and what motivated you to use this loss? Also, have you experimented with other more sophisticated loss functions, e.g., time-frequency domain MSE losses or some other perceptually motivated losses?
>
> **A2**: Thank you for sharing your insight. Training using only SNR loss is a common practice used in numerous prior works [1, 2, 3, 4]. We kept the loss simple and consistent with the prior works to improve the reproducibility and comparability of our work. Therefore, we did not conduct extensive ablation studies on alternative loss functions. Incorporating perceptual losses is indeed a valuable direction for future work to enhance model performance.
>
> [1] J. Huang, X. Wang, and D. Wang, “Multi-level speaker representation for target speaker extraction,” in Proc. ISCA Interspeech, 2021, pp. 1459–1463.
>
> [2] A. Ephrat, I. Mosseri, T. M. Remez, and S. Freeman, “Target conversation extraction: Source separation using turn-taking dynamics,” in Proc. ISCA Interspeech, 2020, pp. 1401–1405.
>
> [3] Y. Xu, M. Yu, C. Yu, and D. Yu, “Listen to extract: Onset-prompted target speaker extraction,” in Proc. ISCA Interspeech, 2022, pp. 2963–2967.
>
> [4] W. Zhou, S. Chen, Y. Liu, and D. Su, “Look once to hear: Target speech hearing with noisy examples,” in Proc. ISCA Interspeech, 2023, pp. 3597–3601.
>
> **Q3**: Will you make the code public if the paper is accepted?
>
> **A3**: Yes. The code and checkpoints will be made public once accepted.

---

> > ### Comment · Reviewer_1MM1 · 2025-08-03
> > **Rebuttal Comment by Reviewer 1MM1**
> >
> > I thank the authors for providing the additional experimental results on naturally recorded real-world audio mixtures. The corresponding subjective evaluation is helpful and further supports the effectiveness of the proposed method. I also agree with reviewer anep's opinion to move some key experiments in the appendices to the main text, especially the results on real-world audio mixtures (Appendix K). From my perspective, as this is a paper that aims to address challenges encountered in realistic settings, highlighting the advantages of the proposed method in such scenarios is important for the reader to appreciate the merits and contributions.
> >
> > As my questions and concerns have been addressed by the authors' rebuttal, and after reading their responses to other reviews, I am now more confident in keeping my positive view of the paper.

---

> > > ### Author Response · Authors · 2025-08-04
> > >
> > > Thank you for your positive comments on our paper. We are happy that our responses addressed your questions! Due to the page limit, the model performance on real-world recordings were placed in the appendix in the initial submission. In the revised version, we will move these key results into the main text to demonstrate the applicability of our model in realistic settings.

---

### Official Review · Reviewer_iQ1f · 2025-07-02

**Clarity:** 2
**Significance:** 2
**Originality:** 3
**Rating:** 4
**Confidence:** 5

**Summary:**

The paper addresses monaural target speaker extraction using only noisy enrollment audio. Instead of requiring clean clips, the authors propose using both Positive (target speaking) and Negative (target silent) segments from noisy enrollments. An attention-based Siamese Encoder Fusion Module generates target-speaker embeddings to guide a TF-GridNet-based Extraction Fusion Module. The two-stage training first distills embeddings from clean speech, then trains the extraction network, leading to faster convergence. The method achieves state-of-the-art results in this setting.

**Questions:**

See Weaknesses for details.

**Ethical Concerns:**

["NO or VERY MINOR ethics concerns only"]

**Final Justification:**

This paper proposes a negative sampling method for target speaker extraction. The experiments demonstrate its effectiveness. However, in my opinion, negative sampling is not new and incremental in 2025. I therefore recommend Borderline Accept, but I will also not be surprised if the paper does not get accepted.

**Limitations:**

Yes.

**Quality:**

2

**Strengths And Weaknesses:**

Strengths:
1. The paper proposes a novel enrollment strategy using paired positive and negative enrollments for addressing monaural target speaker extraction with noisy enrollment.
2. The paper demonstrates robust experiments and comparisons to baseline models.
3. The authors use thorough evaluation on their dataset using objective metrics (SNR, SI-SNR, STOI, DNSMOS) on both monaural and binaural audio.
4. The paper has shown extensive ablation studies on using two-stage training strategy and different fusion strategies to encode the target speaker’s characteristics.

Weaknesses:
1. I think it is important to report performance when a clean enrollment is available. Without this, we cannot judge whether the noisy strategy matches or degrades relative to the ideal case.
2. In my view, comparing only to SpeakerBeam for the monaural task is too weak. Including recent SOTA discriminative and generative TSE methods (e.g. USEF-TSE, SoloAudio) would give a fairer picture of the gains.
3. The current metrics (SNR, SI-SNR, STOI, DNSMOS) omit key measures of intelligibility (WER or CER) and perceptual quality (e.g. PESQ). I think adding these would strengthen the assessment.
4. Objective gains do not always reflect perceptual quality. I would like to see human listening tests or a comparison demo to confirm that improvements hold up for end users.
5. Some notation in Section 3.3 (encoder fusion) is dense. A short pseudo-code snippet or an extra diagram could help readers grasp the self-attention comparison more easily.

---

> ### Author Rebuttal · Authors · 2025-07-29
>
> **Q1**: I think it is important to report performance when a clean enrollment is available. Without this, we cannot judge whether the noisy strategy matches or degrades relative to the ideal case.
>
> **A1**: We reported our model performance when clean enrollments are available in Appendix J in the original supplementary material. Our model outperforms the TSE model trained on clean enrollment when extracting from the mixtures of three or more speakers, but shows lower performance when extracting from the mixtures of two speakers. In addition, we evaluate our model and the baseline models' performance when using noisy single speaker enrollments. Please see the response to reviewer anep Q2 for the result.
>
> **Q2**: Including recent SOTA discriminative and generative TSE methods (e.g. USEF-TSE, SoloAudio) would give a fairer picture of the gains.
>
> **A2**: Thank you for your suggestion. We did not focus our comparison with the SOTA TCE methods because of the difference in problem formulation. Since the enrollments contain multiple speakers, SOTA TSE models trained on clean enrollment cannot perform accurate extraction without additional information to distinguish target speakers from the interfering speakers in the noisy enrollment.
>
> To verify this, we additionally evaluate the USEF-TFGridnet and SoloAudio under the proposed task. As shown in the table below, both model performance degrade significantly under our problem formulation, highlighting the increased difficulty of our task and demonstrating the effectiveness of our proposed model. In particular, we notice that the SoloAudio model trained to perform target sound extraction struggles to distinguish the subtle difference between different speakers' characteristics. As a result, it often produces outputs with large silent regions, leading to notably low SI-SNRi scores.
>
> | Model | Metric | Extract from 2 spk., 2 spk. in Enroll | Extract from 2 spk., 3 spk. in Enroll | Extract from 2 spk., 4 spk. in Enroll |  Extract from 3 spk., 2 spk. in Enroll | Extract from 3 spk., 3 spk. in Enroll | Extract from 3 spk., 4 spk. in Enroll |
> |-|-|-|-|-|-|-|-|
> | USEF-TFGridnet | SNRi | 3.42 $\pm$ 3.43 | 3.45 $\pm$ 3.58 | 3.30 $\pm$ 3.50 | 4.31 $\pm$ 2.52 | 4.15 $\pm$ 2.47 | 4.23 $\pm$ 2.58 |
> | | SI-SNRi | -0.03 $\pm$ 5.97 | -0.03 $\pm$ 6.42    | -0.14 $\pm$ 6.00 | 0.29 $\pm$ 3.38 | 0.11 $\pm$ 3.36 | 0.09 $\pm$ 3.24 |
> | | PESQ | 1.52 $\pm$ 0.49 | 1.54 $\pm$ 0.49 | 1.52 $\pm$ 0.49 | 1.32 $\pm$ 0.38 | 1.32 $\pm$ 0.37 | 1.31 $\pm$ 0.36 |
> | | STOI | 0.43 $\pm$ 0.17 | 0.43 $\pm$ 0.18 | 0.43 $\pm$ 0.17 | 0.36 $\pm$ 0.11 | 0.35 $\pm$ 0.11 | 0.36 $\pm$ 0.11 |
> | | DNSMOS | 1.37 $\pm$ 0.45 | 1.37 $\pm$ 0.45 | 1.36 $\pm$ 0.45 | 1.35 $\pm$ 0.41 | 1.33 $\pm$ 0.41 | 1.34 $\pm$ 0.42 |
> | | WER | 0.66 $\pm$ 0.33 | 0.66 $\pm$ 0.36 | 0.68 $\pm$ 0.32 | 0.85 $\pm$ 0.26 | 0.86 $\pm$ 0.21 | 0.86 $\pm$ 0.20 |
> | SoloAudio | SNRi | 1.55 $\pm$ 2.26 | 1.39 $\pm$ 2.17 | 1.57 $\pm$ 2.35 | 3.27 $\pm$ 2.41 | 3.45 $\pm$ 2.79 | 3.59 $\pm$ 2.58 |
> | | SI-SNRi | -47.05 $\pm$ 11.04  | -46.40 $\pm$ 12.13  | -46.46 $\pm$ 10.13  | -46.93 $\pm$ 12.77  | -44.46 $\pm$ 11.98  | -45.41 $\pm$ 12.03  |
> | | PESQ | 0.85 $\pm$ 0.90 | 0.60 $\pm$ 0.69 | 0.71 $\pm$ 0.85 | 0.65 $\pm$ 0.76 | 0.69 $\pm$ 0.72 | 0.78 $\pm$ 0.88 |
> | | STOI | 0.06 $\pm$ 0.09 | 0.07 $\pm$ 0.09 | 0.06 $\pm$ 0.09 | 0.04 $\pm$ 0.08 | 0.05 $\pm$ 0.09 | 0.05 $\pm$ 0.08 |
> | | DNSMOS | 1.57 $\pm$ 0.26 | 1.59 $\pm$ 0.24 | 1.58 $\pm$ 0.25 | 1.55 $\pm$ 0.22 | 1.51 $\pm$ 0.26 | 1.54 $\pm$ 0.25 |
> | | WER | 0.96 $\pm$ 0.03 | 0.97 $\pm$ 0.02 | 0.98 $\pm$ 0.01 | 0.98 $\pm$ 0.02 | 0.98 $\pm$ 0.04 | 0.99 $\pm$ 0.04 |
>
>
> **Q3**: The current metrics (SNR, SI-SNR, STOI, DNSMOS) omit key measures of intelligibility (WER or CER) and perceptual quality (e.g. PESQ). I think adding these would strengthen the assessment.
>
> **A3**: Thank you for your suggestion on additional metrics.  We include further evaluations of our models and baseline methods using PESQ and WER in the table below. Our monaural model outperforms all the baseline methods in both metrics. However, as mentioned in the Conclusion and Limitation section, the generated audio still contains artifacts, which negatively impact the PESQ and WER performance. Since the binaural model is trained to preserve the reverberation characteristics of the target speaker, and reverberation can negatively impact PESQ and WER, we report the binaural models' results here solely for completeness.
>
> | Model | Metric | Extract from 2 spk., 2 spk. in Enroll | Extract from 2 spk., 3 spk. in Enroll | Extract from 2 spk., 4 spk. in Enroll |  Extract from 3 spk., 2 spk. in Enroll | Extract from 3 spk., 3 spk. in Enroll | Extract from 3 spk., 4 spk. in Enroll |
> |-|-|-|-|-|-|-|-|
> | Ours (Monaural) | PESQ | **2.07 $\pm$ 0.34** | **2.06 $\pm$ 0.36** | **2.05 $\pm$ 0.36** | **1.79 $\pm$ 0.33** | **1.78 $\pm$ 0.34** | **1.75 $\pm$ 0.37** |
> | | WER | **0.42 $\pm$ 0.35** | **0.43 $\pm$ 0.28** | **0.45 $\pm$ 0.28** | **0.61 $\pm$ 0.28** | **0.61 $\pm$ 0.28** | **0.62 $\pm$ 0.28** |
> | Ours (FiLM Fusion)   | PESQ | 1.84 $\pm$ 0.32 | 1.83 $\pm$ 0.34 | 1.84 $\pm$ 0.33 | 1.57 $\pm$ 0.31 | 1.56 $\pm$ 0.32 | 1.54 $\pm$ 0.35 |
> | | WER | 0.54 $\pm$ 0.37 | 0.54 $\pm$ 0.29 | 0.54 $\pm$ 0.28 | 0.73 $\pm$ 0.24 | 0.72 $\pm$ 0.25 | 0.73 $\pm$ 0.24 |
> | TCE | PESQ   | 1.91 $\pm$ 0.34 | 1.80 $\pm$ 0.41 | 1.73 $\pm$ 0.46 | 1.91 $\pm$ 0.34 | 1.53 $\pm$ 0.39 | 1.48 $\pm$ 0.40 |
> | | WER | 0.73 $\pm$ 0.16 | 0.76 $\pm$ 0.24 | 0.77 $\pm$ 0.24 | 0.73 $\pm$ 0.25 | 0.88 $\pm$ 0.21 | 0.88 $\pm$ 0.17 |
> | USEF-TFGridnet | PESQ | 1.52 $\pm$ 0.49 | 1.54 $\pm$ 0.49 | 1.52 $\pm$ 0.49 | 1.32 $\pm$ 0.38 | 1.32 $\pm$ 0.37 | 1.31 $\pm$ 0.36 |
> | | WER | 0.66 $\pm$ 0.33 | 0.66 $\pm$ 0.36 | 0.68 $\pm$ 0.32 | 0.85 $\pm$ 0.26 | 0.86 $\pm$ 0.21 | 0.86 $\pm$ 0.20 |
>
> | Model | Metric | Extract from 2 spk., 2 spk. in Enroll | Extract from 2 spk., 3 spk. in Enroll | Extract from 2 spk., 4 spk. in Enroll |  Extract from 3 spk., 2 spk. in Enroll | Extract from 3 spk., 3 spk. in Enroll | Extract from 3 spk., 4 spk. in Enroll |
> |-|-|-|-|-|-|-|-|
> | Ours (Binaural) | PESQ | 2.24$\pm$ 0.47 | 2.22 $\pm$ 0.47 | 2.25 $\pm$ 0.47 | 1.85 $\pm$ 0.44 | 1.85 $\pm$ 0.42 | 1.87 $\pm$ 0.43 |
> | | WER | 0.45 $\pm$ 0.48 | 0.45 $\pm$ 0.34 | 0.44 $\pm$ 0.35 | 0.63 $\pm$ 0.33 | 0.64 $\pm$ 0.43 | 0.64 $\pm$ 0.42 |
> | LookOnceToHear| PESQ   | 2.28 $\pm$ 0.47 | 2.25 $\pm$ 0.52 | 2.28 $\pm$ 0.45 | 1.87 $\pm$ 0.48 | 1.88 $\pm$ 0.44 | 1.90 $\pm$ 0.47 |
> | | WER | 0.45 $\pm$ 0.41 | 0.44 $\pm$ 0.35 | 0.47 $\pm$ 0.68 | 0.66 $\pm$ 0.36 | 0.63 $\pm$ 0.33 | 0.64 $\pm$ 0.33 |
>
>
> **Q4**: Objective gains do not always reflect perceptual quality. I would like to see human listening tests or a comparison demo to confirm that improvements hold up for end users.
>
> **A4**:
> We provided five audio examples each for the successful and failure cases of our model in the supplementary materials.
>
> Additionally, we evaluate the Mean Opinion Score (MOS) of different models using naturally recorded audio mixtures sourced from Freesound.org and the VoxConverse dataset. These sounds are real-world audio mixtures captured in pubs, metro stations, urban areas, and city council meeting. Please see the response to reviewer 1MM1 Q1 for the results.
>
> **Q5**: Some notation in Section 3.3 (encoder fusion) is dense. A short pseudo-code snippet or an extra diagram could help readers grasp the self-attention comparison more easily.
>
> **A5**: Thank you for your suggestion. We provide the pseudo-code of the Encoder Fusion Module below. Two segmentation embeddings, $S_{pos}, S_{neg}$, are first element-wise added to the input embeddings $E_{pos}, E_{neg}$ to allow the model to distinguish which enrollment each embedding originates from. The resulting two embeddings are concatenated along the temporal dimension and passed through two Full-band Self-attention layers.
>
> The self-attention calculation between the embeddings of different Positive Enrollment frames allows the model to identify the Positive Interferers, who remain silent in some of the Positive Enrollment frames, and exclude these speakers in the extracted embedding. Similarly, attention between the Positive and Negative Enrollments embeddings enables the model to identify and exclude Negative Interferers, (i.e., speakers present only in the negative enrollments), from the output embedding.
>
> During Stage 1 training, we simulate the enrollments such that the number of Positive and Negative Interferers varies between 0 and 2. Distilling the encoder to produce an embedding that retains only the target speaker’s characteristics under these enrollments input encourages the model to exclude both types of interferers in the extracted speaker embedding.
>
> ```python
> # Input:
> #     E_pos: Positive Enrollment embedding, shape [T_pos, D]
> #     E_neg: Negative Enrollment embedding, shape [T_neg, D]
> # Learnable parameters:
> #     S_pos: segmentation embedding for E_pos, shape [1, D]
> #     S_neg segmentation embedding for E_neg, shape [1, D]
> #     M: 2 Full-band Self-attention layers
> Encoder Fusion Module:
>   # Elementwise add segmentation embedding via broadcasting
>   E_pos = E_pos + S_pos # shape [T_pos, D]
>   E_neg = E_neg + S_neg # shape [T_neg, D]
>
>   # Concatenate along the temporal dimension
>   E_concat = [E_pos, E_neg] # shape [T_pos + T_neg, D]
>
>   # Apply two Full-band Self-attention calculation
>   E_concat = M(E_concat) # shape [T_pos + T_neg, D]
>
>   # Truncate embeddings to match the teacher model's embedding shape
>   output = E_concat[:T_pos] # shape [T_pos, D]
> ```

---

### Official Review · Reviewer_anep · 2025-07-04

**Clarity:** 2
**Significance:** 3
**Originality:** 3
**Rating:** 4
**Confidence:** 4

**Summary:**

This paper proposes a novel target speaker extraction (TSE) model capable of utilizing mixture enrollments with multiple speakers for effective extraction of the target speaker's voice. The enrollment feature is first enhanced by extracting positive and negative enrollment representations from segments with and without the target speaker respectively. An self-attention mechanism is then employed to filter out interference speakers from the enrollment feature, resulting in a fused enrollment feature focused on the target speaker. It is finally integrated into the TSE model via cross-attention to extract the target speaker's voice. The proposed model is mostly based on the TF-GridNet model architecture, and the core contribution is the ability of utilizing mixture enrollments for effective TSE. Experiments are conducted on Librispeech and WSJ0 (Appendix) based simulation datasets as well as a real-world dataset based on VoxConverse, which validates the efficacy of the proposed model, especially in scenarios with multiple speakers in the enrollment.

**Questions:**

1. When the input signal is binaural (as described in Section 4.1), it is unclear whether the proposed model is capable of modeling the spatial information between two microphone channels, as Section 3 only discusses monaural scenarios.

2. Table 1 primarily evaluates the extraction performance when there are multiple speakers in the enrollment, which matches the motivation of this paper. However, it is still necessary to report the performance with single-speaker noisy enrollments, as this constitutes a large percentage of real-world scenarios. In Appendix J, it is shown that the proposed model underperforms TD-SpeakerBeam with single-speaker clean enrollments. But it is still unclear whether the same tendency will be observed for noisy enrollments.

3. While line 323 claims that the model “correctly extracts the target speaker” based on the improved performance shown in Figure 5, it is not explicitly verified. In particular, the target confusion problem [R1] is commonly observed in target speaker extraction systems and can be analyzed quantitatively. Therefore, it is highly recommended that the authors present in-depth analysis on the accuracy of target speaker extraction, which can be more difficult in the context of mixture enrollments as discussed in this paper.

[R1] Z. Zhao, D. Yang, R. Gu, H. Zhang, and Y. Zou, “Target confusion in end-to-end speaker extraction: Analysis and approaches,” in Proc. ISCA Interspeech, 2022, pp. 5333–5337.

4. There are two major limitations of the experiments:
    1. Only the non-intrusive DNSMOS metric is used for the evaluation on real-world speech data in Appendix K. It is nice to see evaluation on real-world data. However, as reported by various prior studies, solely using an NN-based non-intrusive metric for evaluation can be risky as they may not reflect the true speech quality due to domain mismatch. Additional evaluation metrics such as word error rate (WER), speaker similarity, or even MOS based on subjective listening tests should be additionally reported for more comprehensive assessment.
    2. There is no demo page or subjective listening to subjectively validate the performance of the proposed method. For example, the artifacts mentioned in line 365 cannot be easily understood without an example.

**Ethical Concerns:**

["NO or VERY MINOR ethics concerns only"]

**Limitations:**

The potential negative societal impact of this work is not discussed in Appendix N. It is recommended that the authors include a discussion on the potential misuse of the proposed method and how they can avoided. For example, the proposed method could raise privacy concerns if used to extract the voice of a target speaker without their consent.

**Paper Formatting Concerns:**

1. The full name of SI-SNR (scale-invariant SNR) should be explicitly mentioned, and a reference should be added.
2. According to Appendix C, the SpeakerBeam model used in Section 4 is actually TD-SpeakerBeam, a time-domain variant of the original frequency-domain version. So the reference [5] should be updated.
3. The sentence between lines 285 and 290 is difficult to follow and should be rephrased.
4. "Since its very challenging" -> "Since it is very challenging"
5. The expression “more optimal” sounds weird as “optimal” already means “the best”.
6. "Film" -> "FiLM"
7. The conference names are missing in most references.

**Quality:**

3

**Strengths And Weaknesses:**

Strengths:
1. The problem addressed by the proposed method is important and novel, which is highly related to real-world applications.
2. The proposed model is compact and performant, achieving better performance in complex scenarios with mixture enrollments with only 1.88 M parameters, which is smaller than baseline models such as TCE and LookOnceToHear.
3. Extensive experimental validation and ablation studies are provided on both simulated and real-world datasets, validating the model's robustness and generalizability.

Weaknesses:
1. Some technical details are missing, such as the definition and implementation of the learnable segmentation embeddings in the Encoder Fusion Module.
2. The organization of the paper could be improved by moving some key experiments (e.g., Appendix K) to the main text to better highlight the effectiveness of the proposed method.
3. Some claims lack a solid experimental validation, such as line 323, which should be rephrased or supported by additional experiments.
4. The writing needs improvement, as some sentences are overly complex or even grammatically incorrect, leading to poor readability.
5. More advanced TSE baselines, such as X-TF-GridNet, instead of the outdated SpeakerBeam model, should be compared in the experiments to better assess the performance of the proposed method.

---

> ### Author Rebuttal · Authors · 2025-07-29
>
> **Q1**: When the input signal is binaural (as described in Section 4.1), it is unclear whether the proposed model is capable of modeling the spatial information between two microphone channels, as Section 3 only discusses monaural scenarios.
>
> **A1**: Thank you for raising this question. The difference between the monaural and binaural audio architectures lies solely in the input. The binaural model takes in 4 channels input (stacked real and imaginary components from both channels) and the monaural model takes in only 2 channels (stacked real and imaginary component). By stacking the two channels' STFT, the binaural model can capture the inter-channel temporal differences between the two channels, which encode the directional information of the target speaker since the target speaker speaks at 90 degree azimuth angle in the enrollment. Both model architectures do not differ after the first convolution layer.
>
> To verify that our model leverages spatial information for speaker extraction, we evaluate the binaural model by randomly varying the azimuth angle of the target Speaker in the Positive Enrollment. As shown in the table below, this results in a significant drop in performance. This indicates that our binaural model leverages the spatial information (target speaker direction) to perform enrollment.
>
> | Model | Metric    | Extract from 2 speakers, Random Azimuth   | Extract from 2 speakers, Azimuth = 90 degree | Extract from 3 speakers, Random Azimuth | Extract from 3 speakers, Azimuth = 90 degree |
> |-|-|-|-|-|-|
> | Ours (Binaural) | SNRi      | 6.91 $\pm$ 3.56 | 9.60 $\pm$ 3.57        | 7.96 $\pm$ 3.46 | 9.78 $\pm$ 3.23 |
> | | SI-SNRi   | 2.22 $\pm$ 8.01 | 7.84 $\pm$ 5.20 | 1.81 $\pm$ 8.03 | 6.72 $\pm$ 5.08 |
> | | PESQ      | 1.80 $\pm$ 0.62 | 2.22 $\pm$ 0.47 | 1.54 $\pm$ 0.54 | 1.85 $\pm$ 0.42 |
> | | STOI      | 0.58 $\pm$ 0.22 | 0.73 $\pm$ 0.16 | 0.47 $\pm$ 0.23 | 0.61 $\pm$ 0.18 |
> | | DNSMOS    | 1.68 $\pm$ 0.52 | 1.80 $\pm$ 0.59 | 1.53 $\pm$ 0.46 | 1.60 $\pm$ 0.49 |
> | | WER       | 0.65 $\pm$ 0.40 | 0.45 $\pm$ 0.34 | 0.78 $\pm$ 0.33 | 0.64 $\pm$ 0.43 |
>
>
>
>
> **Q2**: Model performance with single-speaker noisy enrollments.
>
> **A2**: Thank you for your suggestion on additional experiments. We re-evaluate our model and two baseline models using noisy single speaker enrollments. The noise in the enrollment is from the WHAM! dataset and set at 0 SNR level. The table below shows the model performance when extracting from the mixture of two or three speakers, TSE models trained with clean enrollments show significant performance decrease under noisy single speaker enrollments. In comparison, our model consistently outperforms the baselines across all metrics except DNSMOS, demonstrating greater robustness to enrollment noise.
>
>
> | Model          | Metric       | Extract from 2 speakers | Extract from 3 speakers |
> |-|-|-|-|
> | Ours (Monaural) | SNRi         | **10.19 $\pm$ 2.59**         | **10.48 $\pm$ 2.46**         |
> | | SI-SNRi | **9.12 $\pm$ 3.50**          | **8.50 $\pm$ 3.63**          |
> | | PESQ | **2.08 $\pm$ 0.32**          | **1.81 $\pm$ 0.33**          |
> | | STOI | **0.76 $\pm$ 0.10**          | **0.67 $\pm$ 0.12**          |
> | | DNSMOS | 2.15 $\pm$ 0.37          | 1.93 $\pm$ 0.37          |
> | | WER | **0.43 $\pm$ 0.29**          | **0.61 $\pm$ 0.35**          |
> | USEF-TFGridnet | SNRi         | 3.54 $\pm$ 3.45          | 4.29 $\pm$ 2.46          |
> | | SI-SNRi      | 0.50 $\pm$ 5.58          | 0.56 $\pm$ 2.85          |
> | | PESQ         | 1.52 $\pm$ 0.48          | 1.33 $\pm$ 0.36          |
> | | STOI         | 0.47 $\pm$ 0.17          | 0.38 $\pm$ 0.11          |
> | | DNSMOS       | 1.35 $\pm$ 0.44          | 1.29 $\pm$ 0.37          |
> | | WER          | 0.64 $\pm$ 0.33          | 0.84 $\pm$ 0.25          |
> | TD-SpeakerBeam    | SNRi         | 8.88 $\pm$ 5.05          | 7.21 $\pm$ 3.44          |
> | | SI-SNRi      | 4.88 $\pm$ 10.82         | 1.77 $\pm$ 8.24          |
> | | PESQ         | 1.81 $\pm$ 0.71          | 1.32 $\pm$ 0.57          |
> | | STOI         | 0.62 $\pm$ 0.23          | 0.47 $\pm$ 0.19          |
> | | DNSMOS       | **2.50 $\pm$ 0.43**      | **2.33 $\pm$ 0.41**      |
> | | WER          | 0.63 $\pm$ 0.31          | 0.83 $\pm$ 0.26          |
>
>
> **Q3**: Line 323: Explanation of "correctly extracts the target speaker" and analysis of the target speaker confusion problem.
>
> **A3**： Thank you for your suggestion. By "correctly extracts the target speaker", we mean the extracted audio achieving a higher SI-SNR with respect to the target speaker's voice than with respect to the interferer's speech. If the model mistakenly encodes the interfering speaker in the enrollment, it should instead show a high SI-SNR for the interfering speaker. As shown in Figures 4 and 5 in the original paper, the extracted speech has less than -10 Si-SNRi with respect to the interfering speaker's speech, verifying that the interfering speakers' voice is effectively suppressed.
>
> To further investigate whether our model suffers from the target speaker confusion problem, we follow the experimental setup proposed by Zhao et al. [1]. Specifically, we construct 5000 test samples where both the audio mixture and the enrollment contain two speakers, and share the same interfering speaker. Let $P_{pred}$ be the enrollment pair in the sample, where speaker A is the target speaker and speaker B is the interfering speaker. After performing extraction, we identify the enrollment pairs that result in the extracted audio being more similar (in terms of SNR) to the interfering speaker B's speech than to the target speaker A's. We refer to these samples as target confusion samples. For each target confusion sample, we construct two additional enrollment pairs: $P_{tgt}$ and $P_{interfer}$. In $P_{tgt}$, same as the target confusion sample, speaker A is the target and speaker B is the interferer. In $P_{interfer}$, speaker B is the target and speaker A is the interferer. The table below summarises the role of speaker A and speaker B in the audio mixture and each enrollment pair.
>
> | | Audio Mixture | $P_{pred}$ | $P_{tgt}$ | $P_{interfer}$ |
> |-|-|-|-|-|
> | Target Speaker | A | A | A | B |
> | Interfering Speaker | B | B | B | A |
>
> To verify if the model mistakenly encode the interfering speaker as the target speaker and the interfering speaker, we flatten and normalize the embeddings extracted by our model from each enrollment pair, obtaining $E(P_{pred})$, $E(P_{tgt}), E(P_{interfer})$, and compute the cosine similarity between $E(P_{pred})$ and $E(P_{tgt})$, as well as between $E(P_{pred})$ and $E(P_{interfer})$.
>
> Out of the 5000 tested samples, 74 of them are the target confusion samples. 34 samples out of the 74 target confusion samples (45.9%) have enrollment embedding closer to the interfering speaker than to the target speaker (i.e. 34 of the samples have $cos(E(P_{pred}), E(P_{tgt})) < cos(E(P_{pred}), E(P_{interfer}))$). These findings are consistent with those reported in the target speaker confusion study [1], where 45.1% of target confusion samples showed the extracted embedding being closer to the interferer. This means that significant percentage of target confusion samples have encoder embeddings being more close to the interfering speaker. However, it is important to emphasise that only 74 out of the 5000 samples (1.48%) showed evidence of target speaker confusion in our evaluation, suggesting that this is a rare occurrence and does not pose a significant issue for our model's overall performance.
>
> [1] Z. Zhao, D. Yang, R. Gu, H. Zhang, and Y. Zou, “Target confusion in end-to-end speaker extraction: Analysis and approaches,” in Proc. ISCA Interspeech, 2022, pp. 5333–5337.
>
> **Q4**: Additional evaluation metrics and baseline comparison on real-world speech data.
>
> **A4**: We additionally evaluate our model with WER and PESQ on the synthetic data. Please see the response to reviewer iQ1f Q3 for the results. In addition, since we do not have access to the ground truth of the extracted audio in audio mixtures recorded in the real-world, we focus our evaluation on the Mean Opinion Score for the real-world speech data. Please see the response to reviewer 1MM1 Q1 for the results.
>
> **Q5**: The artefacts mentioned in line 365 cannot be easily understood without an example.
>
> **A5**: We provided 5 audio examples for each of successful and failure cases of our model in the original supplementary file. We will implement a demo page to present the extraction results on both synthetic and real-world audio mixtures.
>
> **Q6**: More advanced TSE baselines, such as X-TF-GridNet, instead of the outdated SpeakerBeam model, should be compared in the experiments to better assess the performance of the proposed method.
>
> **A6**: Thank you for your suggestion. As USEF-TFGridnet[2] reports higher performance in comparison to the X-TFGridnet in their paper, we compare with USEF-TFGridnet and SoloAudio. Please see the response to reviewer iQ1f Q2 for the results.
>
> [2] B. Zeng and M. Li, “USEF‑TSE: Universal Speaker Embedding Free Target Speaker Extraction,” IEEE Trans. Audio, Speech and Language Processing, 2025, pp. 2110–2124.
>
> **Paper Formatting Concerns and Clarifying Technical Details**
> Thank you for raising these concerns. We will refine the grammar and explanation in the corresponding paragraphs in the revised version. In particular, we will include pseudo-code of the Encoder Fusion module in the response to reviewer iQ1f Q5. We will also discuss the potential negative social impact mentioned.

---

> > ### Comment · Area_Chair_nkU7 · 2025-08-07
> > **Request anep to actively participate in the discussion period**
> >
> > Dear Reviewer anep:
> >
> >  The authors have responded to several of you comment. For example, they provided new results to show that the model  utilizes spatial cues for speaker extraction. Specifically, they assess the binaural model by randomly altering the azimuth angle of the target speaker in the Positive Enrollment.  What is your opinion on their new insights.
> >
> >  The authors also provided 5 audio examples for each of successful and failure cases of our model in the original supplementary file to address your Q5. Have you inspected those examples?
> >
> > Moreover, performance on single-speaker noisy enrollments is provided in the rebuttal.
> >
> >  Thank you
> >
> > Regards
> > AC

---

### Note · Authors · 2025-08-11

We thank all the reviewers for their constructive feedback and suggestions for additional experiments. We summarise our contributions as follow:
- We tackle the problem of target speaker extraction (TSE) in noisy environments, where enrollments inevitably contain interfering speakers and background noise, through comparing a pair of noisy positive and negative audio enrollments.
- We propose a two-stage training paradigm to mitigate the slow convergence issue in noisy-enrollment TSE, validated through ablation studies.
- Extensive experiments on both simulated and real-world datasets demonstrate the robustness and generalizability of our model.

Reviewers recognized that our novel TSE pipeline tackles a realistic yet under-explored challenge, as reflected in originality scores above 3 in their initial evaluations. They also mentioned in the paper strengths that the extensive experiments validate the model's robustness and generalizability. The concerns raised by reviewers focused on requests for model architecture details and additional evaluation results, which we addressed accordingly in the rebuttal.

In particular, we provided further explanation and pseudo-code for the Encoder Fusion Module, which resolved reviewer iQ1f's concern. We also included additional evaluation metrics, comparisons against stronger baselines, and a user study on real-world audio mixtures. Nonetheless, these additional experiments further demonstrate our model's robustness and generalizability, and support our contributions listed above. No further questions were raised after these clarifications.

Including these additional clarifications and experiments in the revised paper will require only minor refinements to the explanations in the relevant sections and the addition of further results in the supplementary material, without any major changes to the core content of the paper. In conclusion, the reviewers responded positively to the paper’s novelty and effectiveness, and the concerns they raised have been addressed, with no further issues remaining for discussion.

---

### Decision · Program_Chairs · 2025-09-17

**Decision:**

Accept (poster)

**Comment:**

In this work, the authors tackle the problem of extracting a target speaker's voice from a mixture when clean reference audio is not available. They propose a method that learns speaker characteristics from noisy enrollment signals, which include both speech and silence segments.

Key strengths include:
(i) A novel enrollment strategy that leverages paired positive and negative samples to address monaural target speaker extraction under noisy conditions;
(ii) Comprehensive experiments demonstrating that the proposed approach outperforms prior methods, achieving over 2.1 dB improvement in SI-SNRi and significantly faster training convergence.

Main weaknesses:
The presentation quality could be improved. Conceptually, the idea is akin to negative sampling, which is not entirely new and may be considered incremental. The experiments presented during the rebuttal phase were understandably absent from the initial submission and should be incorporated into the revised manuscript before publication.

The rebuttal phase was productive, with both authors and nearly all reviewers actively engaging in discussion. Unfortunately, Reviewer anep, despite providing a solid initial review and my requestes to participate in the discussion, did not participate in the discussion. Nevertheless, their comments were addressed by the authors. Overall, reviewers’ feedback primarily focused on the need for additional experiments to strengthen the empirical validation. The authors conducted these experiments as requested. In reaching my recommendation, I placed less weight on Reviewer anep's review due to their lack of engagement during the discussion phase. While I agree with Reviewer iQ1f’s observation that the approach is not fully novel, I believe the work merits acceptance as a poster presentation. Notably, some reviewers increased their scores following the discussion.